# Uranium-stibinidiide, -stibinidene, and -stibido multiple bonds and uranium-nitride formation from multimetallic diuranium-distibene-mediated dinitrogen cleavage

Rebecca F. Sheppard [1], Kevin Dollberg[2], Nick Michel[2], John A. Seed[1], Ashley J. Wooles [1], Jingzhen Du [1,3] ✉, Carsten von Hänisch[2] ✉ & Stephen T. Liddle [1] ✉

Although uranium-nitrogen multiple bonding is well developed, there are far fewer uranium-phosphorus and -arsenic multiple bonds, and none for antimony, even in spectroscopic scenarios. Here, we report straightforward syntheses of uranium-stibido, -stibinidiide, -distibene, and -stibinidene derivatives containing single, double, and pseudo-triple bond interactions. Quantum chemical calculations suggest that these uranium-antimony multiple bonds are more covalent than thorium-antimony congeners, due to superior spatial and energy matching of uranium and antimony frontier orbitals, but comparison to isostructural uranium-phosphorus and -arsenic analogues suggests that for uranium moving from phosphorus to arsenic to antimony the spatial overlap term reduces but the orbital energy matching improves. Reduction of the distibene complex results in loss of the antimony-component and multimetallic activation and cleavage of dinitrogen to nitride. This constitutes an uncommon mode of reactivity for uranium that is co-facilitated by the distibene and potassium ions.

In recent years the chemistry of actinide (An) -ligand multiple bonding[1–5], in particular with Group 15 elements[6,7], has become relatively well developed. However, whilst there has been significant progress with An-N-ligand multiple bonding[1–7], there are far fewer examples of An-multiple bonding to P-, As-, and Sb-ligands, and the numbers dwindle rapidly as Group 15 is descended[8]. Indeed, Group 15 ions become increasingly metallic and electropositive in nature as the group is descended; on the one hand this is attractive, because it constitutes a natural nexus of An-ligand and -metal multiple bonding, on the other hand it intrinsically presents the increased challenge of bonding increasingly large, electropositive An and Group 15 ions together.

Following on from our work on An-P and -As multiple bonding[9–15], An-Sb multiple bonds were disclosed in 2024[16], where we found that the parent antimonide reagent [K(18C6)(THF)SbH$_2$] (**1K**, 18C6 = 18-crown-6 ether)[17] reacts with a range of Th-reagents to produce discrete bridging stibido (Th-Sb=Th), stibinidiide (Th-Sb(H)-Th), stibinidiide (Th=Sb(H)K(18C6)), stibinidene (Th=Sb(H)), distibene (Th(μ-η$^2$:η$^2$-Sb$_2$) Th), and tetrameric stibido ([Th≡SbK$_2$]$_4$) structural motifs[16]. Subsequently, utilising [Cs(18C6)$_2$][SbH$_2$][17] instead of **1K** produced tri-An undeca-antimontriide Zintl clusters[18] and the distibene (Sb$_2$$^{2-}$) complex could be reduced to its radical trianion Sb$_2$$^{•3-}$ form[19]. Prior to those An-Sb complexes, even An-Sb single bonds remained relatively rare[20–23]. Having secured Th-Sb multiple bonds, our attention turned to U-Sb

[1]Department of Chemistry and Centre for Radiochemistry Research, The University of Manchester, Manchester, UK. [2]Fachbereich Chemie and Marburg Center for Quantum Materials and Sustainable Technologies (mar.quest), Philipps-Universität Marburg, Marburg, Germany. [3]Present address: College of Chemistry, Zhengzhou University, Zhengzhou, China. ✉e-mail: jingzhendu@zzu.edu.cn; haenisch@chemie.uni-marburg.de; steve.liddle@manchester.ac.uk

multiple bonds, which in principle are more challenging to secure due to the potentially facile and deleterious range of redox decomposition pathways available to U[18]. Indeed, U-Sb multiple bonds have no precedent, even in spectroscopic scenarios. Given the already observed variance of reactivity of the parent antimonide reagent, dependent on the alkali metal, we examined the reactivity of [Na(15C5)SbH$_2$] (**1Na**, 15C5 = 15-crown-5 ether)[17] and **1K** towards the U-reagents [U{N(CH$_2$CH$_2$NSiPr$^i_3$)$_2$(CH$_2$CH$_2$NSiPr$^i_2$CHMeCH$_2$)}][24] (**2U**), [U{N(CH$_2$CH$_2$NSiCy$_3$)$_2$(CH$_2$CH$_2$NSiCy$_2$[CHCH$_2$CH$_2$CH$_2$CH])}][14] (**3U**), and [U(Tren$^{TIPS}$)(THF)][BPh$_4$] (**4U**, Tren$^{TIPS}$ = {N(CH$_2$CH$_2$NSiPr$^i_3$)$_3$}$^{3-}$; THF = tetrahydrofuran)[9].

Here, utilising **1Na** and **1K** in protonolysis and salt elimination reactions with **2U**–**4U** we report the synthesis, isolation, and characterisation of U-stibido, -stibinidiide, -distibene, and -stibinidene derivatives. These complexes present U-Sb bonds spanning single, double, and pseudo-triple bond interactions, and provide a wealth of structural motifs from a small range of precursors. Quantum chemical calculations suggest that the U-Sb bonds reported here are more covalent than Th-Sb congeners, due to better spatial and energy matching of the U and Sb frontier orbitals, but comparison to isostructural U-P and U-As analogues suggests that for U on moving from P, to As, to Sb the spatial overlap term reduces but the orbital energy matching improves. Lastly, reduction of the distibene complex results in loss of the Sb-component and activation and complete cleavage of

N$_2$ to N$^{3-}$, which remains an uncommon mode of reactivity for U-complexes[25–33].

## Results

### Synthesis and characterisation of 5UNa, 5UK, and 5UK'

Addition of **1Na** or **1K**[17] to solutions of the uranium(IV) Tren$^{TIPS}$-cyclometallate complex **2U**[24] produced brown solids following work-up (Fig. 1a). Reactions proceeded to 50% conversion for 1:1 reactions so the syntheses were optimised to 1:2 ratios. Following work-up the red-black stibido complexes [M(L)][{U(Tren$^{TIPS}$)}$_2$(μ-Sb)] (M = Na, L = (15C5) (OEt$_2$), **5UNa**; M = K, L = 18C6, **5UK**) were isolated in 15 and 46% crystalline yields. Providing a further variant, addition of 2.2.2-cryptand to **5UK** (Fig. 1a), displaces the 18C6 from K producing red-black crystals of [K(2.2.2-cryptand)][{U(Tren$^{TIPS}$)}$_2$(μ-Sb)] (**5UK'**) in 72% yield.

The $^1$H and $^{29}$Si{$^1$H} Nuclear Magnetic Resonance (NMR) spectra of **5UNa**, **5UK**, and **5UK'** indicate symmetric and common uranium components in solution (Supplementary Figs. 29–37), and suggest that the uranium(IV) oxidation state has been retained in the products[34]. The infrared (IR) spectra of **5UNa**, **5UK**, and **5UK'** (Supplementary Figs. 63–65) do not exhibit any absorptions attributable to Sb-H linkages, consistent with both Sb-H linkages in **1Na/1K** being consumed by protonolysis. The formulations of **5UNa**, **5UK**, and **5UK'** imply the presence of U-Sb multiple bonds, however facile sample decomposition prevented the acquisition of Raman spectroscopic data.

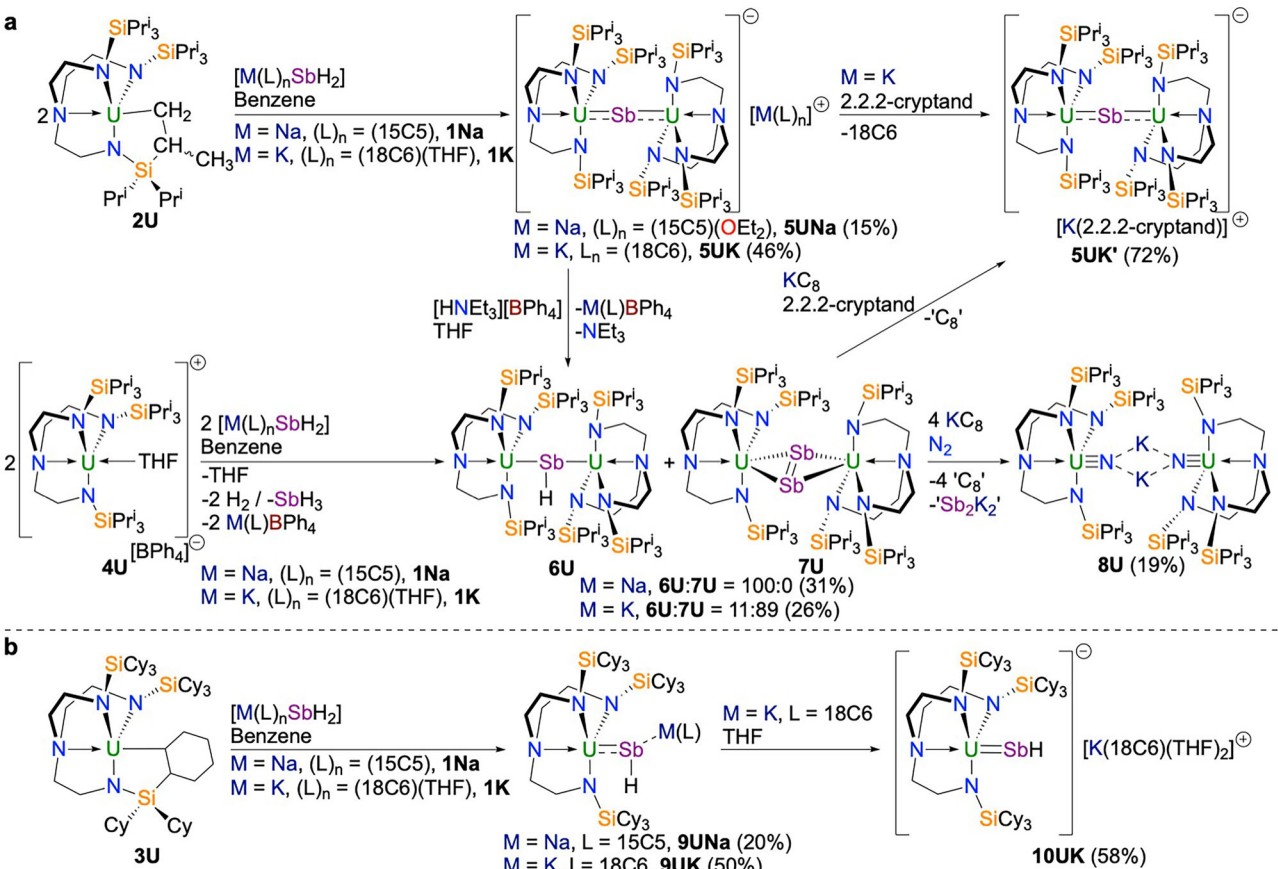

**Fig. 1 | Synthesis of 5UNa, 5UK, 5UK', 6U, 7U, 9UNa, 9UK, 10UK. a** Complex **2U** reacts with **1Na** (15C5 = 15-crown-5 ether) or **1K** (18C6 = 18-crown-6 ether) to afford stibido complexes **5UNa** or **5UK**, respectively, by protonolysis. Complex **5UK** can be converted to **5UK'** by the addition of 2.2.2-cryptand which displaces the 18C6 from the K-ion. Complex **4U** reacts with **1Na** or **1K** by salt elimination and dehydrocoupling to afford either stibinidiide **6U** or a mixture of **6U/7U** (distibene), respectively. Complex **6U** can also be isolated by protonolysis of **5UK** with a proton source. Reduction of the **6U/7U** mixture by KC$_8$ in the presence of 2.2.2-cryptand affords the stibido complex **5UK'**. Reduction of the **6U/7U** mix by KC$_8$ without co-ligands results in N$_2$ cleavage to nitrides with loss of the Sb-portion. **b** Complex **3U** reacts with **1Na** or **1K** by protonolysis to afford the stibinidiides **9UNa** or **9UK**, respectively. Addition of THF to **9UK** produces the separated ion pair stibinidene complex **10UK**.

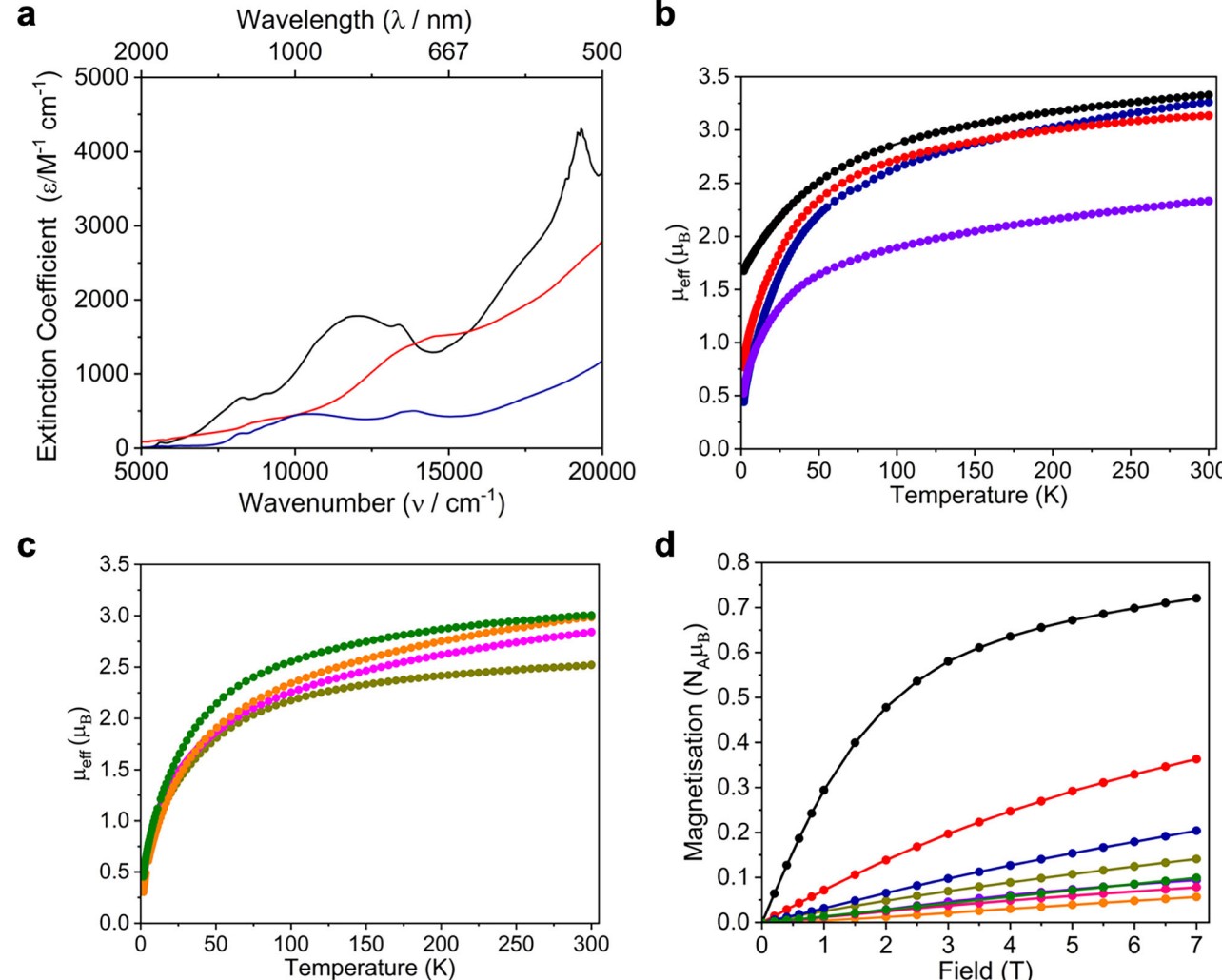

**Fig. 2 | Ultraviolet/visible/near-infrared (UV/Vis/NIR) spectra of 5UK, 6U:7U, and 10UK and variable-temperature superconducting quantum interference device (SQUID) magnetometry data for 5UNa, 5UK, 5UK', 6U, 6U:7U, 9UNa, 9UK, and 10UK. a** Ultraviolet/visible/near-infrared (UV/Vis/NIR) spectra of **5UK** (black line), **6U:7U** (red line), and **10UK** (dark blue line). The solutions were 10, 10, and 25 mM in THF, respectively. **b** effective magnetic moment ($\mu_{eff}$, $\mu_B$) vs temperature (K) data for **5UNa** (dark blue), **5UK** (red), **5UK'** (purple), and **6U** (black). **c** effective magnetic moment ($\mu_{eff}$, $\mu_B$) vs temperature (K) data for **6U:7U** (dark yellow), **9UNa** (pink), **9UK** (orange), and **10UK** (green). **d** Magnetisation ($N_A\mu_B$) vs field (T) at 1.8 K for **5UNa** (dark blue), **5UK** (red), **5UK'** (purple), **6U** (black), **6U:7U** (dark yellow), **9UNa** (pink), **9UK** (orange), and **10UK** (green). For all complexes the data are per U ion. Lines are a guide to the eye only.

The ultraviolet/visible/near-infrared (UV/Vis/NIR) spectra of **5UNa, 5UK**, and **5UK'**, Fig. 2a and Supplementary Figs. 81–83, are all very similar, and when taken together with the similar NMR data for the uranium components in solution suggests the presence of fully separated ion pairs in solution and hence a common [{U(Tren$^{TIPS}$)}$_2$($\mu$-Sb)]$^-$ anion for all three complexes. The UV/Vis/NIR spectra all exhibit strong and broad absorptions across their visible regions, consistent with the red-black colours of **5UNa, 5UK**, and **5UK'**. Of particular note in the UV/Vis/NIR spectra of **5UNa, 5UK**, and **5UK'** are broad absorptions in the range 10000–14000 cm$^{-1}$ ($\varepsilon$ ~ 1900 M$^{-1}$ cm$^{-1}$) that are red-shifted ($\Delta$ ~ −5000 cm$^{-1}$) and more intense ($\Delta$ ~ 1700 M$^{-1}$ cm$^{-1}$) than the analogous absorption for [K(18C6)][{Th(Tren$^{TIPS}$)}$_2$($\mu$-Sb)] (**5ThK**)[16]. For **5ThK**, that feature was found by TD-DFT to be the result of charge transfer from a filled stibido $\pi$-orbital to vacant Th 6d/5f hybrid orbitals. We thus propose that for **5UNa, 5UK**, and **5UK'** the broad absorptions are similarly stibido $\pi$-orbital to vacant U 6d/5f orbital charge transfer transitions. The energy and extinction coefficient values of the **5UNa, 5UK**, and **5UK'** absorptions in question suggests, for U compared to Th, improved energy matching of these transitions

and more orbital mixing, implying greater covalency of the U=Sb=U compared to Th=Sb=Th linkages.

The magnetic responses of **5UNa, 5UK**, and **5UK'** were probed using variable-temperature superconducting quantum interference device (SQUID) magnetometry (Fig. 2a). The data for **5UNa** and **5UK** are similar with effective magnetic moments of 4.61 and 4.43 $\mu_B$ per complex at 300 K (3.26 and 3.13 $\mu_B$ per U ion respectively) that decrease smoothly to 0.62 and 1.08 $\mu_B$ at 1.8 K (0.44 and 0.77 $\mu_B$ per U ion respectively), respectively. In contrast, the effective magnetic moment data for **5UK'** are 3.30 $\mu_B$ (2.34 $\mu_B$ per U ion) at 300 K and 0.74 $\mu_B$ (0.52 $\mu_B$ per U ion) at 1.8 K. Although there is some variance, subtle changes in packing and bond lengths and angles are well known to modulate magnetic properties[35], but these data are consistent with the presence of uranium(IV) ions and magnetic singlet ground states at low temperature[4,36], which is further supported by the isothermal (1.8 K) magnetisation (*M*) vs field (*H*) data for **5UNa, 5UK**, and **5UK'** (Fig. 2d), that are straight lines and not approaching saturation by the highest available field (7 T). At 7 T and 1.8 K the *M* values span the range 0.19–0.73 N$_A\mu_B$ per U complex (0.1–0.36 N$_A\mu_B$ per U ion), which

indicates poorly isolated magnetic singlet ground states, and hence although the χ vs T plots reveal clear maxima (**5UNa**) or shoulders (**5UK**, and **5UK**) that could be interpreted as Néel temperatures of ~30 K resulting from U···U antiferromagnetic exchange coupling, this is instead attributed to single ion crystal field effects[37].

## Synthesis and characterisation of 6U, 7U, and 8U

We next examined the reactivity of **4U**[9] towards **1Na** and **1K**[17] in order to compare salt elimination reactivity to the protonolysis chemistry of **2U**. As for the reactions discussed earlier, reactions proceeded to 50% conversion for 1:1 reactions so the syntheses were optimised to 2:1 ratios. Whereas the reaction outcome for **2U**, in terms of the U-Sb unit that is formed, is essentially independent of the alkali metal cations in **1Na** and **1K**, for **4U** divergent reaction outcomes were observed. When **1Na** is added to **4U** the red-black stibinidiide [{U(Tren$^{TIPS}$)}$_2$(μ-SbH)] (**6U**) is the sole isolable product, typically isolated in crystalline yields of ~30%, Fig. 1a. We also note that **6U** can be prepared from the reaction of [HNEt$_3$][BPh$_4$] with **5UK** or **5UNa**. By contrast, addition of **1K** to **4U** affords a mixture of **6U** and the distibene [{U(Tren$^{TIPS}$)}$_2$(μ-η$^2$:η$^2$-Sb$_2$)] (**7U**), Fig. 1a, isolated as red-black co-crystals in 26% yield. Single crystal X-ray diffraction (see later) reveals that the **6U:7U** ratio is 11:89, so analysis was carried out on co-crystals of **6U:7U** since **7U** is the major component and fractional co-crystallisation was not feasible.

Pure **6U** is insoluble in arene and ethereal solvents once isolated which precluded its analysis by NMR and UV/Vis/NIR spectroscopies. The $^1$H NMR spectrum of the **6U:7U** mixture is rather broad and uninformative, Supplementary Figs. 38 and 39. However, the $^{29}$Si{$^1$H} NMR spectrum, Supplementary Fig. 40, exhibits only one resonance, with a chemical shift consistent with the presence of U(IV)[34], which is likely that of **7U** with the resonance for **6U** not being observed since it is a minor component.

The IR spectrum of **6U**, Supplementary Fig. 66, exhibits an absorption at ~1620 cm$^{-1}$ which is tentatively assigned as the Sb-H stretch. An analytical frequencies calculation predicts that absorption to occur at 1782 cm$^{-1}$, but in addition to necessary approximations in the calculation phase effects likely operate so the level of agreement is considered to be acceptable. The absorption at ~1620 cm$^{-1}$ is also just visible in the IR spectrum of **6U:7U**, Supplementary Fig. 67, consistent with it being the minor component of that co-crystal. The Raman spectrum of **6U:7U**, Supplementary Fig. 78, exhibits two major features below 250 cm$^{-1}$, each of which appears to be a composite of inelastic scattering bands. The analytical frequencies calculation predicts Sb-Sb stretches at 64 and 219 cm$^{-1}$; whilst the Raman data are not definitive they are consistent with the computed Sb-Sb stretching frequencies.

The UV/Vis/NIR spectrum of **6U:7U**, Fig. 2a, exhibits broad and intense absorptions across the visible region consistent with the red-black colour of **6U:7U**. Noting that **7U** is the dominant component, there are two broad features of interest in the UV/Vis/NIR spectrum of **6U:7U**, at ~14000 (at least two absorptions, ε ~ 1400 M$^{-1}$ cm$^{-1}$) and ~20000 cm$^{-1}$ (ε ~ 2800 M$^{-1}$ cm$^{-1}$). These two features are analogous to similar charge transfer features in the UV/Vis/NIR spectrum of **7Th**[16], but are red shifted ~ −10000 cm$^{-1}$ and much broader than for **7Th**. On the basis of the TD-DFT of **7Th** we assign these two features to Sb$_2$ occupied σ-bond to vacant π$_⊥$* and a composite of occupied π$_⊥$ and π$_=$ bonds to vacant π$_⊥$* and U 5f/6d hybrid orbitals.

The magnetic response of **6U:7U** is 3.53 μ$_B$ per complex at 300 K (2.52 μ$_B$ per U ion) and this falls smoothly to 0.65 μ$_B$ at 1.8 K (0.47 μ$_B$ per U ion), Fig. 2c. The isothermal (1.8 K) M vs H data present a straight line not approaching saturation with a value of 0.28 N$_A$μ$_B$ per complex (0.14 N$_A$μ$_B$ per U ion) ion at 7 T, Fig. 2d. These data are consistent with the presence of uranium(IV) that is a magnetic singlet at low temperature[4], like **5UNa**, **5UK**, and **5UK'**. By contrast to **5UNa**, **5UK**, **5UK'**, and **6U:7U**, the variable-temperature SQUID magnetometry data for **6U** are quite distinct. The effective magnetic moment of **6U** at 300 K is 4.7 μ$_B$ per molecule (3.33 μ$_B$ per U ion) and this slowly

decreases reaching a value of 2.37 μ$_B$ per molecule (1.67 μ$_B$ per U ion) at 1.8 K, Fig. 3a. The isothermal (1.8 K) M vs H data reveal a value of 1.44 N$_A$μ$_B$ per molecule at 7 T (0.72 N$_A$μ$_B$ per U ion) and the magnetisation plot is distinctly curved and approaching saturation at 7 T, Fig. 2d. This indicates that **6U** is uranium(IV) but has a low temperature doublet or pseudo-doublet ground state (magnetic triplet)[38], likely due to HSb$^{2-}$ being a strong donor, even when bridging, that imposes axial symmetry and hence a pseudosymmetric electronic structure onto the $^3$H$_4$ ground multiplet. We note that **6U** is a neutral complex, whereas strong donor effects would be likely lessened in anions such as in **5UNa**, **5UK**, **5UK'**, and **10UK** (see later) which would explain why those complexes are magnetic singlets at low temperature.

We previously found that reduction of **6Th/7Th** by KC$_8$ produced the tetrameric thorium stibido complex [{Th(Tren$^{TIPS}$)(μ-SbK$_2$)}$_4$][16]. We therefore explored the analogous reaction of **6U/7U** with KC$_8$ (2:1 K:U ratio), under N$_2$, and after work up dark red crystals of the diuranium dinitride complex [{U(Tren$^{TIPS}$)(μ-NK)}$_2$] (**8U**) were isolated in 19% yield, Fig. 1a[39–41]. Unfortunately, we could not identify the fate of the Sb-component, and when the reaction was conducted under argon no tractable products could be identified. By contrast, reduction of [U(Tren$^{TIPS}$)][42] or **4U** by KC$_8$ does not afford **8U**, with only the formation of [U(Tren$^{TIPS}$)][42] or Tren$^{TIPS}$H$_3$ proligand[42] being observed by NMR spectroscopy. Furthermore, we find that reduction of pure **6U** with KC$_8$, with or without 2.2.2-cryptand, forms mainly Tren$^{TIPS}$H$_3$ proligand and [U(Tren$^{TIPS}$)][42]. Moreover, reduction of **6U/7U** with KC$_8$ (1:1 K:U ratio) results in the formation of Tren$^{TIPS}$H$_3$ proligand[42]. The characterisation data for **8U**, including a new single crystal structure of this complex depicted in Supplementary Fig. 6, unambiguously match those of an authentic sample prepared from reduction of [U(Tren$^{TIPS}$)(N$_3$)] by KC$_8$[39]. The implication of the formation of **8U** is that atmospheric N$_2$ has been cleaved, and it is known that alkali metals can perform an important role in activating and cleaving N$_2$ in multi-metallic cooperative reactions[28,30,33,43–48]. We therefore reduced **6U:7U** with KC$_8$ in the presence of 2.2.2-cryptand to sequester the K-ions, and find that under those conditions **8U** is not formed and instead **5UK'** is the sole isolable product.

## Synthesis and characterisation of 9UNa, 9UK, and 10UK

Switching to the sterically demanding Tren$^{TCHS}$-cyclometallate complex **3U**[14], we find that treatment with **1Na** or **1K**[17] effects protonolysis. Reactions were optimal with 1:1 reactant ratios due to the size of the Tren$^{TCHS}$ ligand suppressing 2:1 reactivity. In both cases the alkali metal capped uranium stibinidiide complexes [U(Tren$^{TCHS}$)(μ-SbH)M(L)] (M = Na, L = 15C5, **9UNa**; M = K, L = 18C6, **9UK**) were isolated as red-black crystals in 20 and 50% yields, respectively (Fig. 1b). Addition of THF to **9UK** affords the separated ion pair uranium stibinidene complex [K(18C6)(THF)$_2$][U(Tren$^{TCHS}$)(SbH)] (**10UK**) as red-black crystals in 58% isolated yield (Fig. 1b). The analogous reaction with **9UNa** led to decomposition.

The $^1$H and $^{29}$Si{$^1$H} NMR and UV/Vis/NIR spectra of **9UNa** and **9UK** could not be obtained due to them being insoluble in aromatic and ethereal solvents once isolated. Data could be acquired for **10UK**, Supplementary Figs. 41 and 42, and the $^1$H NMR spectrum of **10UK** is indicative of a symmetric species in solution consistent with **10UK** being a separated ion pair. The $^{29}$Si{$^1$H} NMR spectrum of **10UK** reveals a single resonance in the region expected for a uranium(IV) complex[34].

The IR spectra of **9UNa**, **9UK**, and **10UK**, Supplementary Figs. 69–71, exhibit broad (fwhm ~80–100 cm$^{-1}$) absorptions at 1574, 1654, and 1653 cm$^{-1}$, which are assigned as Sb-H stretches since analytical frequency calculations predict those absorptions to occur at 1777, 1762, and 1606 cm$^{-1}$; a combination of the Sb-H absorptions being broad and measured/computed in different phases accounts for the variance of agreement across **9UNa**, **9UK**, and **10UK**. The analytical frequencies calculations also predict U=Sb stretches at 144/180, 153/190, and 153/208 cm$^{-1}$. The Raman spectra for **9UK** and **10UK**,

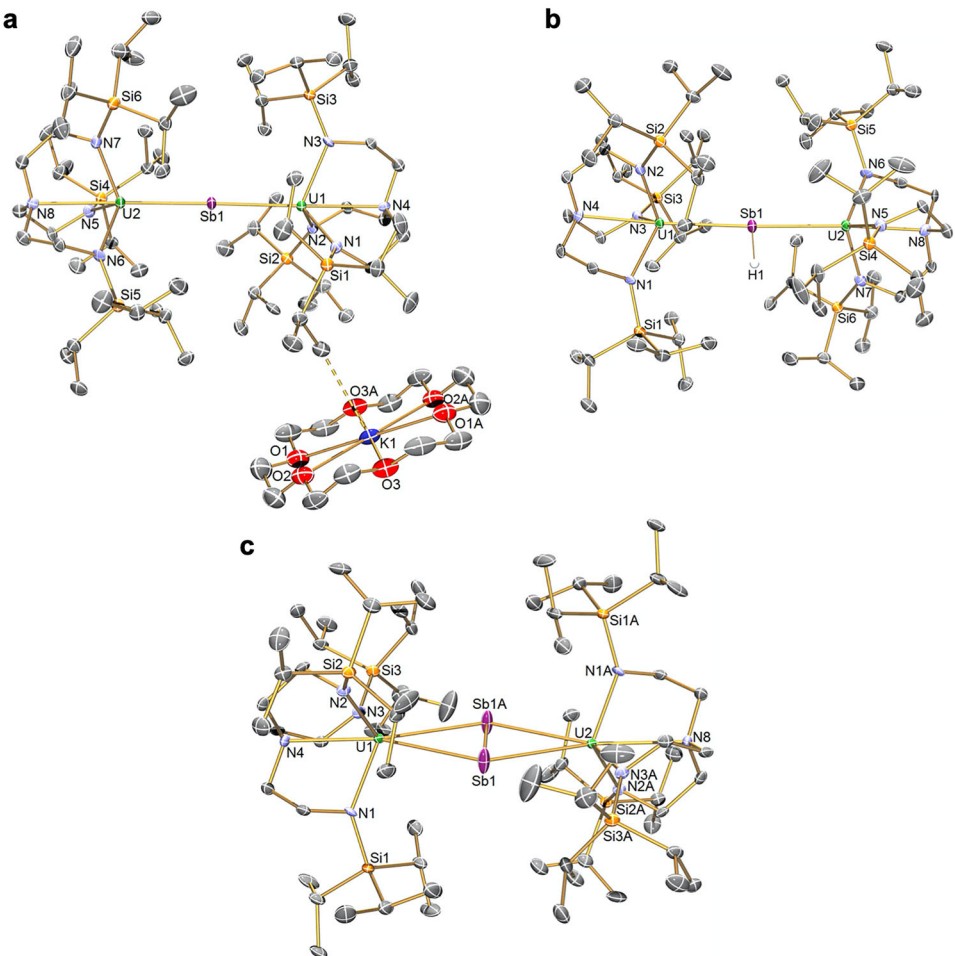

**Fig. 3 | Solid-state structures of 5UK, 6U, and 7U at 150 K. a** Complex **5UK**. **b** Complex **6U**. **c** Complex **7U**. Displacement ellipsoids are at 30%, and non Sb-H hydrogen atoms, lattice solvent, and disordered components are omitted for clarity. The bridging interactions from potassium to neighbouring Pr$^i$ units in **5UK** are omitted for clarity.

Supplementary Figs. 79 and 80, both exhibit two major complex features <250 cm$^{-1}$ that are clearly composed of multiple inelastic scattering bands, so the Raman data are consistent with the computed U=Sb stretching frequencies, but are not definitive.

The UV/Vis/NIR spectrum of **10UK**, Fig. 2a, exhibits, like all the other U-Sb complexes reported here, broad and strong absorptions across the visible range consistent with its red-black colour. The most prominent features are at 8000 (ε ~ 200 M$^{-1}$ cm$^{-1}$), 10500 (ε ~ 480 M$^{-1}$ cm$^{-1}$), 14000 (ε ~ 500 M$^{-1}$ cm$^{-1}$) and 19000 (ε ~ 1000 M$^{-1}$ cm$^{-1}$) cm$^{-1}$, and on the basis of previous TD-DFT calculations on [K(2.2.2-cryptand)][Th(Tren$^{TCHS}$)(SbH)] (**10ThK'**)[16] the latter three likely result from charge transfer from the occupied U=Sb π- and σ-bonds to vacant π* and U 5f/6d hybrid orbitals. The feature at 8000 cm$^{-1}$ is likely a f-f transition, but is it notably large for a f-f transition, since such transitions are usually <50 cm$^{-1}$. This suggests that intensity stealing is operating[49], which implies vibronic coupling through mixing of U and Sb orbitals and hence some covalency in the U=Sb linkage of **10UK**.

The uranium(IV) formulations of **9UNa, 9UK**, and **10UK** were confirmed by variable-temperature SQUID magnetometry, Fig. 2c. The effective magnetic moments of **9UNa, 9UK**, and **10UK** were found to be 2.84, 2.98, and 3.00 μ$_B$ per complex at 300 K, and in each case these values decrease smoothly reaching 0.44, 0.30, and 0.45 μ$_B$ at 1.8 K, respectively. The isothermal (1.8 K) $M$ vs $H$ data for **9UNa, 9UK**, and **10UK** present straight lines that are not approaching saturation, with

$M$ values at 7 T of 0.16, 0.11, and 0.20 N$_A$μ$_B$, respectively, per complex (and hence per U ion) Fig. 2d. These data emphasise that these complexes exhibit typical uranium(IV) behaviour with magnetic singlet ground states at low temperature[4,36,38].

## Solid-state structures

Single crystal X-ray diffraction was used to verify the solid-state formulations of **5UNa, 5UK, 5UK', 6U, 7U, 9UNa, 9UK**, and **10UK**. Where present, Sb-H atoms were identified and refined as described previously[9–12,14,16].

The salient feature of **5UNa, 5UK**, and **5UK'**, Fig. 3a and Supplementary Figs. 2 and 3, is that they all contain a central U=Sb=U stibido unit, with the alkali metal cation components either being separate or at most weakly interacting with Pr$^i$ units. The U-Sb distances are fairly consistent (**5UNa**: 2.9849(3)/2.9962(3), av. 2.991; **5UK**: 2.9691(5)/2.9875(5), av. 2.978; **5UK'** 2.984(2)/3.037(2), av. 3.011 Å), revealing a variance of only 0.03 Å, and the U-Sb-U angles are essentially linear or exactly linear due to crystallographic symmetry (**5UNa**: 179.36(2); **5UK**: 178.22(2); **5UK'**: 180°). The U-Sb distances in **5UNa, 5UK**, and **5UK'** reside within the range of the sum of the single (3.1 Å) and double (2.67 Å) bond covalent radii of U and Sb[50], but are towards the top end of the range reflecting the bridging nature of the stibido ligands in **5UNa, 5UK**, and **5UK'**. The U-Sb distances in **5UNa, 5UK**, and **5UK'** are 0.05–0.08 Å shorter than the analogous Th-Sb distances in previously reported isostructural **5ThK**[16], whereas uranium is usually accepted to

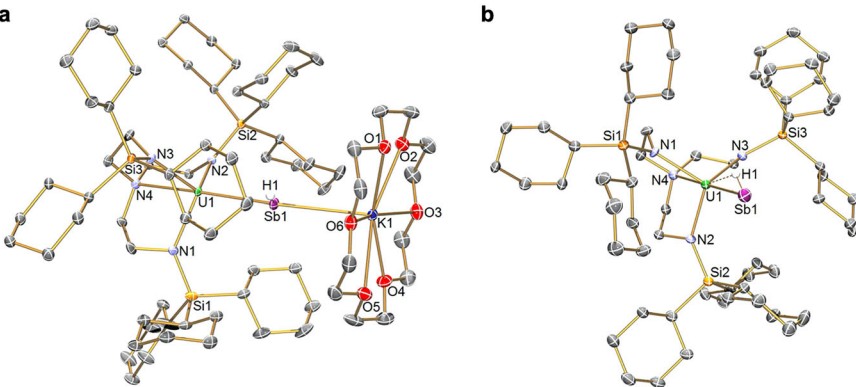

**Fig. 4 | Solid-state structures of 9UK and the anion component of 10UK at 150 and 100 K, respectively. a** Complex **9UK**. **b** Anion component of **10UK**. Displacement ellipsoids are at 30%, and non Sb-H hydrogen atoms, lattice solvent, and disorder components are omitted for clarity.

have a ~0.05 Å smaller ionic radius than thorium, and substantially shorter than the U-Sb single bond distances for [{U(Tren^TIPS)}₃(μ₃-Sb₁₁)] (3.2619(8) to 3.3116(8) Å)[18] and [U(Tren^TIPS){Sb(SiMe₃)₂}] (U-Sb = 3.2089(6) Å)[21]. The U-N_amide (**5UNa**: 2.294(4)−2.3139(3), av. 2.302; **5UK**: 2.280(6)−2.315(7), av. 2.291; **5UK'**: 2.297−2.300, av. 2.299 Å) and U-N_amine (**5UNa**: 2.762(4)/2.772(3), av. 2.767; **5UK**: 2.791(7)/2.816(7), av. 2.804; **5UK'**: 2.741(16)/2.772(13), av. 2.757 Å) distances are typical of uranium(IV)-Tren complexes[51], particularly those where the uranium component is formally anionic.

Complex **6U**, Fig. 3b, crystallises with an essentially T-shaped stibinidiide geometry (U-Sb-U: 177.9(2); U-Sb-H: 89.9(2)/92.1(2)°) and U-Sb distances of 3.0780(5)/3.1445(5) Å, which reflects the bridging nature of the HSb²⁻ unit in **6U**. The U-Sb distances in **6U** are in-line with the sum of the single bond covalent radii of U and Sb (3.1 Å)[50], but are significantly shorter (Δ: 0.08−0.15 Å) than the corresponding Th-Sb distances in previously reported isostructural **6Th**[16]. The U-Sb distances in **6U** are substantially shorter than those of [{U(Tren^TIPS)}₃(μ₃-Sb₁₁)] (3.2619(8) to 3.3116(8) Å)[18] and [U(Tren^TIPS){Sb(SiMe₃)₂}] (U-Sb = 3.2089(6) Å)[21], which can be related to stibinidiide being a dianion and also sterically unencumbered. The U-N_amide (2.223(7)−2.264(6), av. 2.250 Å) and U-N_amine (2.691(6)/2.703(7), av. 2.697 Å) bond lengths are typical of uranium(IV)-Tren complexes[51].

The solid-state structure of **7U**, Fig. 3c, was determined to be an 89:11 co-crystal of **7U:6U** in the crystal examined, and also the Sb₂-unit is disordered over two sites. Nevertheless, the metrical data are reliable and confirm the presence of a distibene, and for **7U** the U-Sb distances (3.289(5)−3.321(4), av. 3.305 Å) are longer than the sum of the single bond covalent radii of U and Sb (3.1 Å)[50], reflecting that the Sb₂²⁻ unit is side-on bridging to both U-ions. The Sb-Sb distances (2.546(10)/2.597(6) Å) reflect the presence of a formal Sb=Sb double bond in **7U** (Sb-Sb single, double, and triple bond covalent radii are 2.80, 2.66, and 2.54 Å, respectively)[19,50]. As above, the U-Sb distances in **7U** are shorter than the analogous Th-Sb distances in previously reported isostructural **7Th**[16], (Δ: ~0.7 Å), but of more interest is that the Sb-Sb distances in **7U** are ~0.07−0.09 Å shorter than in **7Th** (2.640(2)/2.668(8) Å)[16], perhaps indicating more Sb₂ triple bond character in **7U** than **7Th** which could be related to Th-Sb₂-Th being 6d⁰5f⁰ whereas U-Sb₂-U is 6d⁰5f⁴ (see quantum chemical calculations analysis later). The U-N_amide (2.263(4)−2.666(5), av. 2.265 Å) and U-N_amine (2.706(5)/2.706(5), av. 2.706 Å) distances are unexceptional[51].

The salient feature of the solid-state structures of **9UNa** and **9UK**, Fig. 4a and Supplementary Figs. 7 and 8 is the presence of stibinidiide units that bridge U and Na/K atoms, respectively. There are two molecules of **9UNa** in the crystallographic asymmetric unit, one with disorder of the U-Sb-Na unit and one ordered so discussion will focus on the ordered component. The U-Sb(H)-M units adopt distorted T-shaped geometries (U-Sb-Na/K = 163.4(2)/166.6(2); U-Sb-H = 99.2(2)/

97.6(2) Na/K-Sb-H = 84.9(3)/84.4(2)°). Whilst the bridging HSb²⁻ units in **9UNa/9UK** in principle suggest single bond U-Sb and Sb-M interactions, the M ions would be anticipated to be bound electrostatically presenting the opportunity for U=Sb double bond interactions to result. Indeed, for **9UNa** and **9UK** the U-Sb distances of 2.9668(13) and 2.9871(3) Å, respectively, are shorter than the U-Sb distances in **5UNa, 5UK, 5UK'**, and **6U**, and are between the sum of the single and double bond radii of U and Sb (3.1 and 2.67 Å, respectively)[50], suggesting the presence of polar U=Sb double bonding. Indeed, as with earlier complexes reported here the U-Sb distances in **9UNa** and **9UK** are substantially shorter than found for [{U(Tren^TIPS)}₃(μ₃-Sb₁₁)] (3.2619(8) to 3.3116(8) Å)[18] and [U(Tren^TIPS){Sb(SiMe₃)₂}] (U-Sb = 3.2089(6) Å)[21]. The U-Sb distances in **9UNa** and **9UK** are ~0.07−0.09 Å shorter than the analogous Th-Sb distances in previously reported isostructural **9ThK** (Th-Sb: 3.0554(2) Å)[16]. The **9UNa** and **9UK** Sb-M distances of 3.027(7) and 3.4067(9) Å, respectively, are slightly longer than the sum of the single bond covalent radii of Sb/Na and Sb/K (2.95 and 3.36 Å, respectively)[50] consistent with their electrostatic bonding. The Sb-H units in **9UNa** and **9UK** exhibit Sb-H distances of 1.747(2) and 1.749(2) Å, which are slightly longer than the sum of the single bond covalent radii of Sb and H (1.72 Å)[50]. The U⋯H distances in **9UNa** and **9UK** were found to be 3.675(2) and 3.656(5) Å, and the U-Sb-H angles are 97.6(2) and 99.2(7)°; given that the sum of the covalent single bond radii of U and H is 2.03 Å[50] it would seem that any U⋯H interactions are weak at best. The U-N_amide (**9UNa**: 2.287(12)−2.302(10), av. 2.292; **9UK**: 2.283(3)−2.311(3), av. 2.293 Å) and -N_amine (**9UNa**: 2.698(10); **9UK**: 2.709(3) Å) distances are intermediate to those of **6U** and **5UNa, 5UK**, and **5UK'**, reflecting the presence of the HSb²⁻ dianion that is still bridging like in **6U** but evidently bonding more like a terminal linkage.

The separated ion pair formulation, and hence isolation of a terminal parent uranium stibinidene motif with a polar covalent U=Sb double bond is confirmed by the solid-state structure of **10UK**, Fig. 4b and Supplementary Fig. 9. The U-Sb distance of 2.9884(4) Å is statistically indistinguishable from that of **9UK**. We suggest this can be accounted for by the fact that although the U-Sb linkage would, all other things being equal, be anticipated to become shorter on moving from a bridging to terminal motif, the stibinidene component of **10UK** is an anion and hence charge-rich. Based on the very similar U-Sb distances in **9UK** and **10UK** it would appear these two factors essentially cancel each other out. The U-Sb distance is ~0.09 Å shorter than the Th-Sb distance in structurally analogous **10ThK'** (Th-Sb: 3.0729(4) Å)[16]. Interestingly, the Th-Sb bond in **10ThK'** is ~0.018 Å longer than in **9ThK** suggesting that the U-Sb bond of **10UK** is more covalent than the analogous Th-Sb bond in **10ThK'**. The charge-rich nature of **10UK**, and the presence of the terminally bonded HSb²⁻ stibinidene is also reflected by U-N_amide (2.286(4)−2.313(4), av. 2.303 Å) and -N_amine (2.689(5) Å) distances that are similar to the analogous

metrical data in **5UK, 5UK'**, and **6U**. The Sb-H and U···H distances (1.785(4) and 2.418(8) Å) and U-Sb-H angle (36.7(2)°) suggest, given the aforementioned single bond covalent radii of Sb/H and U/H, and noting the shorter Sb-H and longer U···H distances, and less acute U-Sb-H angle in **9UNa** and **9UK**, that the Sb-H single bond is slightly elongated due to a modest U···H interaction in **10UK**. Similar weak interactions have been noted with M=EH (M = U, Th; E = P, As, Sb) bonds previously[9–12,14,16], but usually more for Th than U likely reflecting that Th(IV) is more electron deficient than U(IV), but a Sb-H bond will have more hydridic character than P-H and As-H making the former a better donor than the latter pair.

### Quantum chemical calculations analysis

To probe the nature of the U-Sb linkages reported here, we conducted density functional theory (DFT), Natural Bond Orbital (NBO), Natural Localised Molecular Orbital (NLMO), and Quantum Theory of Atoms in Molecules (QTAIM) calculations, though we did not apply the NBO/NLMO analyses to **7U** due to its four-centre U$_2$Sb$_2$ bonding interactions, Table 1, Supplementary Figs. 96–111 and Supplementary Tables 7–10. We include data for analogous Th-derivatives[16] in Table 1 for comparative purposes. In general the geometry optimised structures satisfactorily match the solid state structures well, and given prior benchmarking studies[15,41,52] we used the same functional (BP86) and basis sets (ZORA TZP) as were utilised in studies of analogous Th-Sb complexes[16] to enable internally consistent comparisons to be made. Discussion will focus on the polar covalent single, double, and pseudo-triple U-Sb bond interactions of **5U** (anion of **5UK**), **6U, 9UNa, 9UK**, and **10U** (anion of **10UK**), however we note in passing that the data for **7U** are consistent with a diuranium(IV)-distibene and hence the central Sb$_2$ unit (Nalewajski-Mrozek bond order of 1.41) contains a Sb=Sb double bond.

The computed charges for U are consistent with uranium(IV)-Tren complexes generally (-2.2–2.8), though we note that those of **9UK**, and **10U** are, being <2, unusually low suggesting significant donation from the coordinated ligands and in particular from the HSb$^{2-}$ dianions. The calculated Sb charges reflects that the stibido in **5U** is a trianion but able to donate to two U-ions whereas the stibinidiide dianion in **6U** is clearly bound quite ionically to two U-centres. In contrast, the computed Sb charges in **9UNa, 9UK**, and **10U** are relatively low, implying substantial donation to the respective U-ions. The computed Nalewajski-Mrozek bond orders align with the expected trends. For example, for **5U** the bond order of 1.86 (av.) is consistent with a polar, bridging U-Sb pseudo-triple bond interaction that in a Lewis description would be represented with a formal covalent double bond, and then as expected in **6U** the bridging HSb$^{2-}$ has a lower U-Sb bond order (1.23 av.) resulting from a covalent σ-bond supplemented by dative π-donation. Complexes **9UNa** and **9UK** both have bond orders reflecting the presence of U=Sb double bonds in those complexes (1.79 and 1.80, respectively) with attenuation of the U=Sb bond orders by electron density being polarised from the U=Sb bonds towards Na and K. Even though the U=Sb distance hardly changes on becoming terminal in **10U**, the absence of a charge-syphoning alkali metal in stibinidene **10U** compared to stibinidiides **9UNa** and **9UK** is reflected by the U=Sb bond order increasing to 2.17 in **10U**, which implies the onset of a U-Sb pseudo-triple bonding interaction. There is clearly a U···H interaction in **10U** (see Energy Decomposition Analysis (EDA) discussion later), which likely involves the electron density in the Sb-H bond, noting that an extreme description of **10U** would be to view it as a protonated stibido. Hence the Sb-H bond can be viewed as an occupied Sb 'lone pair' that would otherwise constitute a weak, additional pseudo π-type bond to U.

The Kohn Sham frontier molecular orbitals (KSMOs) of **5U, 6U, 9UNa, 9UK**, and **10U** all exhibit clear U-Sb interactions, but all contain various intrusions of orbital coefficients, most notably the N-donor centres, as well as the U-Sb interactions of interest. The NBO and

NLMO calculations both return similar bond compositions that echo the KSMOs, except the NLMO approach captures covalent π-bonding contributions that are missed by the NBO method for **5U** and **6U** so we focus our analysis on the NLMO data.

For all of **5U, 6U, 9UNa, 9UK**, and **10U** in each case the U contribution to the σ-bond component is lower than the π-contribution. There is quite a wide range of U and Sb contributions in response to the precise nature and coordination mode of the Sb moiety which deploys 5s-5p hybrids for σ-bonding and π-bonds that are overwhelmingly dominated by 5p character.

For **5U**, Fig. 5, the two U/Sb σ-bonds are 19/80%, respectively, and the π-bonds are 3-centre-2-electron interactions where each U contributes ~22% and the Sb ~52%. The U σ-components exhibit ~11% 7s contributions and the 6d:5f contributions are reasonably balanced but 6d > 5f, whereas the π-bonds are exclusively 6d and 5f with the latter being over twice that of the former.

The effect of protonating the stibido of **5U** to give the stibinidiide in **6U** is reflected in the NLMO data for the latter, Fig. 6. Specifically, the U contributions to the σ- and π-bonds are now ~15 and ~18%, ~4% reduced from **5U**. Nevertheless, the 7s contribution to the U σ-component remains essentially constant and the same pattern of 6d > 5f and 5f»6d contributions in the σ- and π-bonds, respectively, are found.

For **9UNa, 9UK**, and **10U** the NLMO data are particularly revealing, Fig. 7 and Supplementary Figs. 7–10 The U-Sb σ-components are 8, 10, and 24%, respectively. This demonstrates the polarising nature of the Na$^+$ cation in **9UNa**, syphoning electron density from the U=Sb double bond through the σ-framework, which then reduces on moving to the softer and less polarising K$^+$ in **9UK**. Notably, the U contribution in **10U** is the largest of all the U-Sb complexes in this study, consistent with the terminal, doubly bonded nature of the stibinidene. In contrast, to the U-Sb σ-bonds the corresponding π-components are 28, 28, and 37% U character for **9UNa, 9UK**, and **10U**, respectively. This indicates that those components are largely insensitive to the nature of the coordinated alkali metal, and they are the largest of all the U-Sb complexes in this study. Unsurprisingly, the U-Sb π-bond of **10U** exhibits the largest U contribution of all the U-Sb compound reported here, consistent with the presence of a terminal stibinidene unit.

QTAIM analysis of **5U, 6U, 9UNa, 9UK**, and **10U** located U-Sb (3,−1)-bond critical points (BCPs). The U-Sb BCP electron densities (ρ) of 0.03−0.05 and energy (H) terms of ~ −0.01 emphasise that the U-Sb linkages are rather polar and weak. Nevertheless, although QTAIM tends to report small differences it is notable that the U=Sb multiple bonds in **5U, 9UNa, 9UK**, and **10U** have slightly larger ρ values compared to the formal U-Sb single bonds in **6U**. The U-Sb ellipticity (ε) values are also instructive, noting that single and triple bonds exhibit ε values at or close to zero (cylindrical distribution of electron density around the internuclear axis) and double bonds exhibit ε values in the range 0.2−0.5 (asymmetric electron density distribution)[53]. For **5U** the ε values are close to zero (~0.06) in-line with the pseudo-triple bond description from MO theory, whereas for **6U** the corresponding value averages 0.27 reflecting the covalent σ- and dative π-contributions. Likewise, the ε values for **9UNa, 9UK**, and **10U** (0.31, 0.28, and 0.23, respectively) confirm the presence of U=Sb bonds in those complexes.

In order to provide further insight into the nature of actinide-Group 15 bonds we performed EDA calculations, Table 2. Given the variance of formal charges for the complexes reported here, and that the diuranium species are not well suited to fragmentation studies, we focussed on **10U** because it is conveniently partitioned into [U(Tren$^{TIPS}$)]$^+$ and [HSb]$^{2-}$ fragments and the Th analogue (**10Th**) is also known. Furthermore, we have previously reported the U and Th analogues with P and As double bonds[14] giving six isostructural M = EH (M/E = U/P, **10U$^P$**; U/As, **10U$^{As}$**; ThP, **10Th$^P$**; ThAs, **10Th$^{As}$**) complexes to study. When conducting EDA for M = EH linkages, the HE fragments can be partitioned in different charge and configurational states;

**Table 1 | Selected computed DFT and NLMO data for the U-Sb bonds in 5U (anion of 5UK), 6U, 9UNa, 9UK, and 10U (anion of 10UK) and the previously reported Th-analogues**

| Entry[a] | Bond lengths and indices | | | MDCq charges | | NLMO σ-component (%)[g] | | | | NLMO π-component (%)[g] | | | |
|---|---|---|---|---|---|---|---|---|---|---|---|---|---|
| | Expt.[b] | Calc.[c] | BI[d] | M[e] | Sb[f] | M | Sb | M 7s/7p/6d/5f | Sb 5s/5p | M | Sb | M 7s/7p/6d/5f | Sb 5s/5p |
| 5U | 2.9906 | 3.0253 | 1.92 | 2.30 | -0.71 | 19 | 80 | 10/0/47/43 | 47/53 | 22 + 22 | 53 | 0/0/30/70 | 0/100 |
| | 2.9783 | 3.0614 | 1.80 | 2.37 | | 19 | 80 | 12/0/54/34 | 55/45 | 23 + 23 | 50 | 0/0/30/70 | 0/100 |
| | 3.0105 | | | | | | | | | | | | |
| 5Th | 3.0611 | 3.0937 | 1.38 | 2.40 | -0.82 | 15 | 85 | 17/0/61/22 | 49/51 | 12 + 12 | 71 | 0/0/68/32 | 0/100 |
| | 3.0611 | 3.0937 | 1.38 | 2.40 | | 15 | 85 | 17/0/61/22 | 49/51 | 12 + 12 | 71 | 0/0/68/32 | 0/100 |
| 6U | 3.0780 | 3.1318 | 1.24 | 2.69 | -1.15 | 15 | 84 | 10/0/57/33 | 38/62 | 17 + 18 | 62 | 0/0/34/66 | 2/98 |
| | 3.1445 | 3.1457 | 1.22 | 2.76 | | 15 | 84 | 11/0/57/32 | 52/48 | | | | |
| 6Th | 3.2282 | 3.2720 | 0.91 | 2.56 | -1.07 | 13 | 87 | 23/0/56/21 | 45/55 | 11 + 11 | 77 | 0/0/70/30 | 0/100 |
| | 3.2282 | 3.2733 | 0.91 | 2.55 | | 13 | 87 | 23/0/56/21 | 45/55 | | | | |
| 9UNa | 2.9668 | 3.0408 | 1.79 | 2.22 | -0.46 | 8 | 89 | 3/0/42/55 | 24/76 | 28 | 69 | 0/0/27/73 | 4/96 |
| 9UK | 2.9871 | 3.0696 | 1.80 | 1.99 | -0.82 | 10 | 87 | 0/0/36/64 | 9/91 | 28 | 69 | 0/0/25/75 | 1/99 |
| 9ThK | 3.0554 | 3.1461 | 1.32 | 1.92 | -0.82 | 11 | 88 | 15/0/59/26 | 49/51 | 16 | 84 | 0/0/71/29 | 0/100 |
| 10U | 2.9884 | 2.9950 | 2.17 | 1.80 | -0.41 | 24 | 73 | 1/0/34/65 | 16/84 | 37 | 60 | 0/0/25/75 | 0/100 |
| 10Th | 3.0729 | 3.0729 | 1.57 | 1.70 | -0.49 | 16 | 82 | 6/0/67/27 | 32/68 | 22 | 75 | 0/0/76/24 | 0/100 |

[a]Molecules geometry optimised without symmetry constraints using the BP86 GGA functional and basis sets derived from TZP/ZORA all-electron ADF database.
[b]Experimental M-Sb distance (Å, M = U, Th; s.u. given in the main text are not listed here for brevity) from the single-crystal X-ray diffraction structure.
[c]Computed M-Sb distance (Å).
[d]Nalewajski-Mrozek bond indices.
[e]Multipole derived atomic charges on M.
[f]Multipole derived atomic charges on Sb.
[g]Natural localised molecular orbital analysis; these orbitals are by definition 1-electron occupancy in the α-spin manifold, and consequently the sum of M and Sb contributions is not necessarily 100%. See ref. 16 for further details of **5Th**, **6Th**, **9ThK**, and **10Th**.

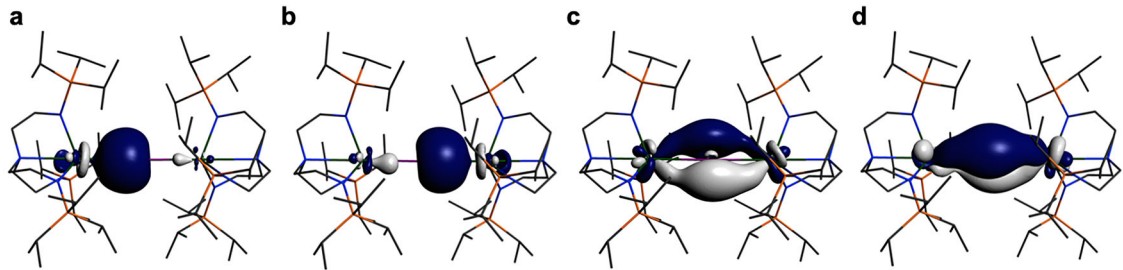

**Fig. 5 | Natural localised molecular orbitals for 5U (%). a** U-Sb σ-bond (19:80 U:Sb). **b** U-Sb σ-bond (19:80 U:Sb). **c** U-Sb-U 3c2e π-bond (22:53:22 U:Sb:U). **d** U-Sb-U 3c2e π-bond (23:50:23 U:Sb:U). Hydrogen atoms are omitted for clarity. The orbitals are rendered at 0.04 a.u. isosurface level. See Table 1 for further details.

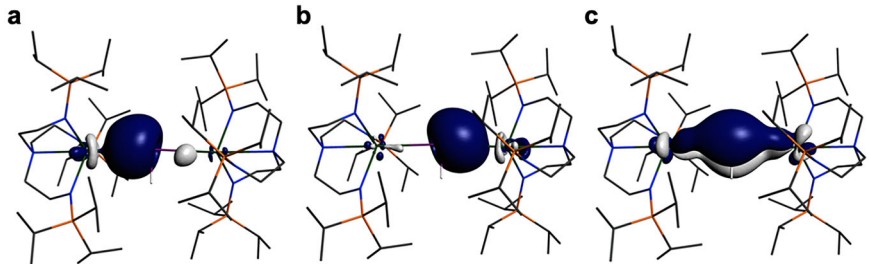

**Fig. 6 | Natural localised molecular orbitals of 6U (%). a** U-Sb σ-bond (15:84 U:Sb). **b** U-Sb σ-bond (15:84 U:Sb). **c** U-Sb-U 3c2e π-bond (17:62:18 U:Sb:U). Non-antimony bound hydrogen atoms are omitted for clarity. The orbitals are rendered at 0.04 a.u. isosurface level. See Table 1 for further details.

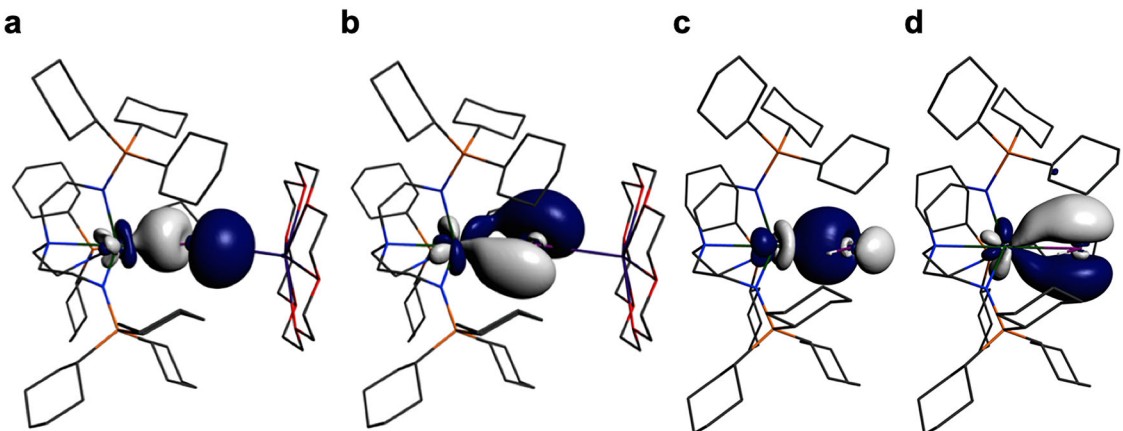

**Fig. 7 | Natural localised molecular orbitals of 9UK and 10U (%). a** U-Sb σ-bond of **9UK** (10:87 U:Sb). **b** U-Sb π-bond of **9UK** (28:69 U:Sb). **c** U-Sb σ-bond of **10U** (24:73 U:Sb). **d** U-Sb π-bond of **10U** (37:60 U:Sb). Non-antimony bound hydrogen atoms are omitted for clarity. The orbitals are rendered at 0.04 a.u. isosurface level. See Table 1 for further details.

however, given the antimonide reagents start as closed shell dianions and the U and Th ions are +4 oxidation state to begin with we focussed on examining [M(Tren$^{TIPS}$)]$^+$ and HE$^{2-}$ fragments, Table 2. The deformation densities for the three principal M=SbH (M = U, Th) interactions are shown in Fig. 8 and the P and As congeners can be found in Supplementary Figs. 112 and 113. Due to the nature of the complexes it was not practicable to correct for basis set superposition errors, preparation energies, or thermal and zero point energies, and so the quoted energies represent an upper bound where the true values would likely be approximately a third of the size[54].

Inspection of Table 2 reveals several notable trends: (i) the Pauli Repulsion of a given UE combination is greater than the corresponding ThE linkage, reflecting that U and Th are 5f$^2$6d$^0$ and 5f$^0$6d$^0$, respectively; (ii) the Pauli Repulsion generally decreases from P to Sb, reflecting reduced destabilising interactions between filled orbitals with longer M = E distances, though it is unexpectedly low for **10U**$^{As}$;

(iii) the Electrostatic Interaction of a given UE unit it greater than the analogous ThE combination, consistent with the more polarising ability of U vs Th, and it decreases from P to Sb consistent with reduced polarising power as Group 15 is descended; (iv) the Orbital Interactions are always greater for U vs Th, suggesting better matching of the former than latter, and decrease from P to Sb as the E valence orbitals become more diffuse; (v) the Orbital Percentages are consistently ~32% indicating that whilst individual contributing components may vary they consistently even out to give meaningful orbital contributions but the majority of the bonding is electrostatic; (vi) the Energy Contributions to the Orbital Interaction are consistently π > σ for U but are σ > π for Th, which may reflect that the U π-bonds are dominated by 5f character but the 6d orbitals of Th may be better suited for direct σ-bonding; (vii) there are EH→M interactions, but these are weakly stabilising or destabilising for U and rather stabilising for Th, as proposed previously and in relation to point (i)[9–12,14].

**Table 2 | Energy decomposition analysis data for the M = EH (M = U, Th; E = P, As, Sb) compounds 10U$^P$, 10U$^{As}$, 10U (anion of 10UK), 10Th$^P$, 10Th$^{As}$, and 10Th$^a$**

| Compound | Pauli repulsion[b] | Electrostatic interaction[c] | Steric interaction[d] | Orbital interaction[e] | Bonding energy[f] | Orbital percentage[g] | M = EH σ-bond[h] | M = EH π-bond[h] | EH→M donation[h] |
|---|---|---|---|---|---|---|---|---|---|
| 10U$^P$ | 346.45 | −440.04 | −93.59 | −217.53 | −311.12 | 33.1 | −64.1 | −64.8 | −0.84 |
| 10U$^{As}$ | 303.03 | −401.06 | −98.03 | −203.20 | −301.23 | 33.6 | −56.4 | −59.7 | +2.27 |
| 10U | 320.34 | −382.36 | −62.02 | −192.86 | −254.88 | 33.5 | −46.1 | −61.8 | +2.23 |
| 10Th$^P$ | 310.25 | −415.27 | −105.02 | −191.70 | −296.72 | 31.6 | −53.7 | −46.2 | −19.4 |
| 10Th$^{As}$ | 276.93 | −383.23 | −106.30 | −179.97 | −286.27 | 31.9 | −51.5 | −38.8 | −21.0 |
| 10$^{Th}$ | 260.88 | −350.25 | −89.37 | −164.10 | −253.48 | 31.9 | −44.8 | −39.7 | −20.2 |

[a]All values in kcal/mol.
[b]Pauli repulsion is the destabilising repulsive interactions between occupied molecular orbitals.
[c]Electrostatic interaction is the classical electrostatic interaction.
[d]The steric interaction is the sum of the Pauli repulsion and electrostatic interaction terms.
[e]The orbital interaction accounts for electron pair bonding, charge transfer, and orbital polarisation.
[f]The bonding energy is the sum of the steric and orbital interaction terms.
[g]The orbital percentage is the percentage of the total attractive forces that is the orbital interaction, i.e. 100 × Orbital Interaction/[Orbital Interaction + Electrostatic Interaction].
[h]Corresponding energy contribution to the Orbital Interaction.

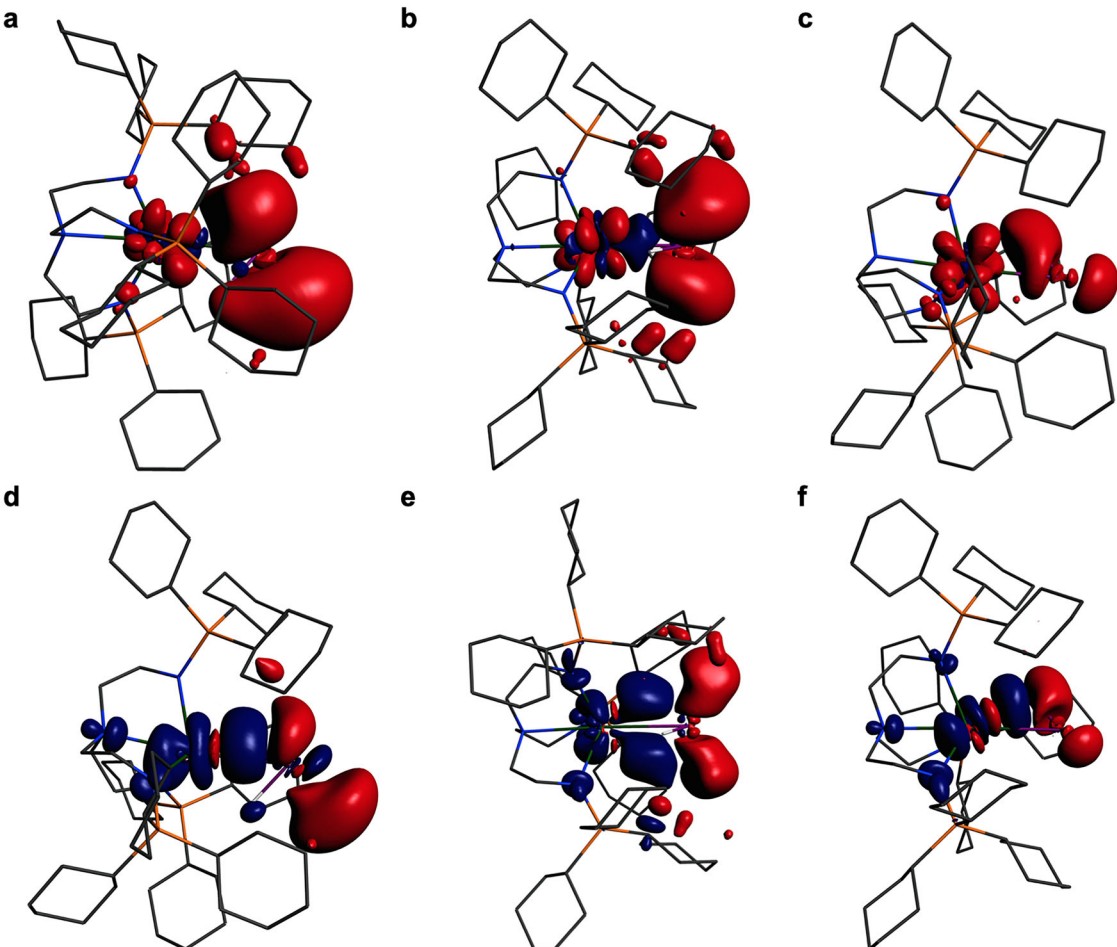

**Fig. 8 | ETS-NOCV deformation density plots highlighting the principal interactions in the M = SbH (M = U, Th) bonds of 10U and 10Th. a** Sb→U σ-bond donation in **10U**. **b** Sb→U π-bond donation in **10U**. **c** Sb-H→U donation in **10U**. **d** Sb→Th σ-bond donation in **10Th**. **e** Sb→Th π-bond donation in **10Th**. **f** Sb-H→Th donation in **10Th**. Charge flow is from red to blue isosurfaces. The deformation density isosurfaces are plotted at 0.001 a.u. and non-Sb bound H-atoms are omitted for clarity. See Table 2 for further details.

## Discussion

The synthesis of **5UNa, 5UK, 5UK', 6U, 7U, 9UNa, 9UK**, and **10UK** highlights the weak and labile Sb-H bonds in **1Na** and **1K** and a tendency towards Sb homo-catenation, usually by dehydrocoupling, that is largely suppressed by the bulky Tren-ligands deployed in this study. Following on from U-P[9,13,14], U-As[10,14], Th-P[11,14,15], Th-As[12,14], and Th-Sb[16,17] analogues, the synthesis of the U-Sb compounds in this work continues to underscore the remarkable level of control in their preparation,

given the large number of bonds that are cleaved and formed in one step, and also the diversity of structural motifs that can be realised from a relatively small pool of U and Sb starting materials when combined with protonolysis and salt elimination reactions. Although U is more redox active than Th, it is notable that the +4 oxidation state of U is maintained in the Sb-reactions reported here, since reactions with the $H_2Sb^-$ anion have been shown before to be easily capable of reducing U[17]. We note that **6U** is the sole isolable product when **1Na** is used, but a mixture of **6U** and **7U** is obtained when **1K** is utilised, which likely reflects that a K-Sb linkage will be more polarised than the corresponding Na-Sb one, and hence for K the $H_2Sb^{1-}$ anion will be more reactive. Complex **7U** can be intuitively considered to be the ultimate product of dehydrocoupling that might be anticipated to be most likely to occur in the presence of highly ionic K-Sb bonds, but **6U** can be thought of as an intermediate, or different reaction path, more likely to occur with Na-Sb linkages that would be more covalent and hence less reactive than K-Sb bonds.

The synthesis of **8U** from **6U/7U** merits specific discussion, since when the analogous reaction is performed with Th the stibido tetramer [{Th(Tren$^{TIPS}$)(μ-SbK$_2$)}$_4$] is formed[16]. Given that U is usually considered to be more covalent in its bonding to ligands than Th, the formation of **8U** instead of "[{U(Tren$^{TIPS}$)(μ-SbK$_2$)}$_4$]" seems initially surprising, since a U≡Sb triple bond would be predicted to be more stable than a Th≡Sb bond; indeed, the uranium-arsendio complex [{U(Tren$^{TIPS}$)(μ-AsK$_2$)}$_4$] has been prepared before[10], albeit by deprotonation of the parent arsenide rather than a redox route. However, we propose that **7U** may present a straightforward reduction path, to give a charge-loaded Sb$_2$-unit, which is credible because P$_2^{\bullet 3-}$ and Sb$_2^{\bullet 3-}$ radical trianion complexes of Tren$^{TIPS}$-U and -Th have been reported[19,55], demonstrating the excellent π-accepting capacity of heavier pnictogens[56]. It may be that elimination of the resulting Sb-component then generates a transient divalent [U(Tren$^{TIPS}$)]$^-$ moiety, which might, in concert with K, be sufficiently reducing to activate and cleave N$_2$. Certainly, our control experiments (see earlier) examining reductions of [U(Tren$^{TIPS}$)][42], **4U**[10], **6U**, and **6U/7U** suggest that pairing U with Sb is important for activating N$_2$ in this context and that the Sb moiety needs to be Sb$_2^{2-}$ rather than HSb$^{2-}$. We surmise that a U(II) oxidation state, at a minimum, would be required, since trivalent [U(Tren$^{TIPS}$)] is stable under N$_2$ for weeks. We note that [U(Tren$^{TIPS}$)] does not seem to be reduced when placed in the presence of excess K, indeed it is prepared in the presence of excess K$_{mirror}$ or KC$_8$[42], and so perhaps the Sb$_2$-unit provides an alternative, energetically more accessible indirect route to reduction of U that is not otherwise normally directly feasible. No tractable products could be obtained when the same reaction is run under argon, but interestingly when run under N$_2$ in the presence of 2.2.2-cryptand **5UK'** is the sole isolable product, suggesting that K plays an important role in multimetallic activation of N$_2$. Indeed, alkali metals have been shown to be intimately involved in N$_2$ reduction[28,30,33,43–48], and cleavage of E$_2^{2-}$ to E$^{3-}$ versus reduction of E$_2^{2-}$ to E$_2^{\bullet 3-}$ (E = P, Sb) has been shown to be dependent on K-reductant being available or sequestered, respectively[16,19,55,57]. When half as many equivalents of KC$_8$ are used only Tren$^{TIPS}$H$_3$ proligand can be isolated, potentially further underscoring the vital role of K-reductant and the likely importance of U(II) to N$_2$ reduction in this context though the Sb$_2^{2-}$ acting as an electron reservoir, for example generating an activated U(III)Sb$_2$U(III) unit, may also play a role. Confirmation of our current working hypothesis awaits a fully elaborated examination of the reduction chemistry of **7U** and [U(Tren$^{TIPS}$)], but nonetheless the observation that reduction of a U$_2$Sb$_2$ complex results in activation and complete cleavage of N$_2$ to N$^{3-}$ is notable given how unusual U-mediated cleavage of N$_2$ to N$^{3-}$ is[25–33].

The USbU stibido motif of **5UNa**, **5UK**, and **5UK'** represents a rare example of a MSbM linkage, joining the small number of d-block[58,59], main group[60–63], and the actinide stibido complex, **5ThK**[16], as well as extending the small range of MEM complexes (M = Th, U; E = P, As) generally[11–13,15,64,65]. Likewise, the μ-Sb-H motifs of **6U**, **9UNa**, and **9UK**

are unusual, with prior examples restricted to gallium derivatives[66,67] and **6Th** and **9ThK**[16], noting that d-block Sb-H complexes exhibit the stibinidiide in a μ$_3$-coordination mode[68]. Prior to **10ThK**[16], and now **10UK**, there were very few structurally authenticated stibinidene complexes, with other examples restricted to [ArBi=SbAr] (Ar = C$_6$H$_2$-2,6-{CH(SiMe$_3$)$_2$}$_2$-4-C(SiMe$_3$)$_3$)[69] and [Ga(SbAr')(BDI)] (Ar' = C$_6$H$_3$-2,6-(C$_6$H$_2$-2,4,6-Me$_3$)$_2$; BDI = HC(CMeNC$_6$H$_3$-2,6-Pr$^i_2$)$_2$)[60]. However, **10ThK** and **10UK** remain notable for only having Sb-H substituents rather than bulky aryl groups and constitute unusually heavy An-element multiple bond combinations[70,71].

The computational analysis, Tables 1 and 2, consistently reveals that like-for-like the U-Sb complexes reported here are more covalent than their isostructural Th-counterparts. Indeed, this is also reflected in experimental observations where apart from various structural metrics and red-shifted/more intense UV/Vis/NIR data that suggest increased U-Sb covalency compared to Th-Sb, it is notable that **10UK** is prepared simply by exposing **9UK** to the monodentate ligand THF to break the contact ion pair into a separated ion pair. In contrast, **9ThK** must be treated with the more powerfully coordinating macrocyclic reagent 2.2.2-cryptand to break the Sb-K interaction. This suggests that the U=Sb double bond is more fully developed than the Th=Sb double bond, and so the K is consequently more weakly bound, and thus easier to remove from the former than the latter, which also is in-line with the computed U=Sb and Th=Sb bond order, NLMO, and EDA data.

The bonding character of the U-Sb linkages reported here are overall dominated by 5f character, as is typical for U, but they also evidence significant and fairly consistent 6d- and 7s-orbital contributions in the σ-bonds, which for the 7s-contributions in particular is unusual. Previously, U-P and U-As σ-bonds[9,10,14] have been found to exhibit only small (<7%) 7s-contributions, but in contrast Th-P[11,14,15], Th-As[12,14], and Th-Sb[16,17] σ-bonds have consistently exhibited significant 7s-character (10–29%). Thus, the 7s-contributions of ~3–12% for the U-Sb σ-bonds in **5UNa**, **5UK**, **5UK'**, **6U**, **7U**, **9UNa**, **9UK**, and **10UK** complement those of their Th-analogues. Where Th comparisons are available for P, As, and Sb we previously noted that for the 7s contribution came at the expense of the 6d, 5f, and 5f character, respectively[11,12,16]. For the U-Sb complexes reported here the 7s contribution also appears to be at the expense of 5f character.

It is clear that on a like-for-like basis U-Sb bonds are more covalent than Th-Sb bonds, and so it seems that U has better spatial and energy matching of the U and Sb frontier orbitals compared to Th/Sb combinations. Although not all U-P/-As/-Sb combinations are available for comparisons, it is instructive to examine the U = EH pnictinidene series. On average, the U = P, U=As, and U=Sb Mayer/Nalewajski bond indices decrease (1.85/2.53, 1.56/2.43, and 1.50/2.26, respectively)[9,10,14], as do the respective QTAIM $\rho$ values (0.08, 0.07, and 0.05) and EDA energies, but conversely the typical U contributions to the U = EH σ-/π-bonds increase (-25/31, -26/36, and -27/38%). These data suggest that for U on moving from P, to As, to Sb the spatial overlap term reduces but the orbital energy matching improves, which is also reflected in the EDA Orbital Percentages (Table 2). For the former component this likely reflects the dominance of 5f character and for the latter likely the 6d and in particular the 7s contributions to the bonding compared to the typical dominance of 6d/5f components in the bonding of U. Hence, the U-Sb bonds reported here seem to benefit more from orbital energy matching rather than overlap driven covalency[72,73].

Synthesising and isolating new chemical bonds is a fundamental endeavour. The complexes reported here introduce a range of new stibido, stibinidiide, distibene, and stibinidene motifs, with U-Sb bonds spanning single, double, and pseudo-triple bond interactions. These unusually heavy An-element combinations, which did not even have precedent in spectroscopic scenarios, constitute a nexus of uranium-ligand and uranium-metal[loid] multiple bonding, providing fundamental insights into heavy element chemical bonding in a relativistic regime.

## Methods

### General

All manipulations were carried out using Schlenk techniques, or an MBraun UniLab glovebox, under an atmosphere of dry $N_2$ or Ar. Solvents were dried by passage through activated alumina towers and degassed before use. All solvents were stored over K-mirrors except for ethers, which were stored over activated 4 Å molecular sieves. D-solvents were distilled from K, degassed by three freeze-pump-thaw cycles and stored under $N_2$. $Cy_2SiCl_2$ and $ClSi^iPr_3$ were distilled from Mg, degassed by three freeze-pump-thaw cycles and stored under $N_2$. The compounds $MO^tBu$ (M = Na, K) and 2.2.2-cryptand was dried under vacuum ($1 \times 10^{-3}$ mbar) for 24 hr before use. $N(CH_2CH_2NH_2)_3$ was distilled prior to use. Elemental potassium was freed from oxides and washed with hexane to remove mineral oil prior to use. Other chemicals were purchased from commercial sources and used as received. Depleted $UO_3$ was supplied by the National Nuclear Laboratory. The compounds $[HNEt_3][BPh_4]$[74], $KC_8$[75], $UCl_4$[76], $KCH_2Ph$[77], $[Na(15C5)SbH_2]$ (**1Na**, 15C5 = 15-crown-5 ether)[17], $[K(18C6)(THF)SbH_2]$ (**1K**, 18C6 = 18-crown-6 ether)[17], $LiCy$[14], $Cy_3SiCl$[14], $Tren^{TIPS}Li_3$[42], $Tren^{TCHS}Li_3$[14], $[U(Tren^{TIPS})Cl]$[42], $[U(Tren^{TCHS})Cl]$[14], $[U(Tren^{TIPS})]$[42], $[U\{N(CH_2CH_2NSiPr^i_3)_2(CH_2CH_2SiPr^i_2CHMeCH_2)\}]$ (**2U**)[24], $[U\{N(CH_2\text{-}CH_2NSiCy_3)_2(CH_2CH_2NSiCy_2[CHCH_2CH_2CH_2CH_2CH])\}]$ (**3U**)[14], and $[U(Tren^{TIPS})(THF)][BPh_4]$ (**4U**, $Tren^{TIPS} = N(CH_2CH_2NSiPr^i_3)_3$)[9] were prepared by the modified procedures described below.

Single crystals were examined on a Rigaku XtalLAB Synergy-S diffractometer, equipped with a HyPix 6000HE photon counting pixel array detector with mirror-monochromated MoKα (λ = 0.71073 Å) or CuKα (λ = 1.54184 Å) radiation. Intensities were integrated from a sphere of data recorded on narrow (0.5°) frames by ω rotation. Cell parameters were refined from the observed positions of all strong reflections in each data set. Gaussian grid face-indexed absorption corrections with a beam profile correction were applied. The structures were solved by dual methods using SHELXT[78] and all non-hydrogen atoms were refined by full-matrix least-squares on all unique $F^2$ values with anisotropic displacement parameters with exceptions noted in the respective cif files. Except where noted hydrogen atoms were refined with constrained geometries and riding thermal parameters; $U_{iso}(H)$ was set at 1.2 (1.5 for methyl groups) times $U_{eq}$ of the parent atom. Using a previously established method[9–12,14,16], for Sb-H hydrogen atom placements the difference map was examined for the likely location of the hydride, and then the selected residual density position was included in a model that was geometry optimised by DFT calculations. The resulting computed Sb-H distance and relevant angles were then used to complete the crystallographic refinement using bond length and angle restraints. In general, the DFT-derived positions were very close to the original Q-peak locations. The largest features in final difference syntheses were close to heavy atoms; this results in cif-check alerts but they are clearly related to absorption effects and are of no chemical significance. CrysAlisPro was used for control and integration[79], and SHELXL[80], Olex2[81], and Platon[82] were employed for structure refinement. ORTEP-3 and POV-Ray were employed for molecular graphics[83,84]. $^1H$, $^{11}B\{^1H\}$, $^{13}C\{^1H\}$, and $^{29}Si\{^1H\}$ spectra were recorded on either a Bruker 400 MHz spectrometer operating at 400, 128.4, 100.6, and 79 MHz, respectively; or, for $^1H$, $^{11}B\{^1H\}$, and $^{29}Si\{^1H\}$, recorded on a JEOL JNM-ECZ 400 MHz spectrometer operating at 399.78, 128.27, and 79.42 MHz, respectively. Chemical shifts and are quoted in ppm, relative to trimethylsilane ($^1H$, $^{13}C$, and $^{29}Si$) or $BF_3.OEt_2$ ($^{11}B$). ATR-IR spectra were recorded on a Bruker Alpha spectrometer with a Platinum-ATR module in a glovebox. Raman spectra were recorded on a Horiba XploRA Plus Raman microscope using a 532 nm laser with a power of 1.5 mW. The power was adjusted using a power filter for each complex to inhibit sample decomposition. UV/Vis/NIR spectra were recorded on a Perkin Elmer LAMBDA™ 1050 spectrometer. Data were collected in a 1 mm path-length cuvette loaded in a glovebox and were run versus the appropriate solvent. Variable-

temperature magnetic moment data were recorded in an applied direct current (DC) field of 0.1 or 0.5 Tesla on a Quantum Design MPMS3 superconducting quantum interference device magnetometer using recrystallised powdered samples. Measurements were performed in dc scan mode using 40 mm scan length and 6 s scan time. Samples were carefully checked for purity and data reproducibility between independently prepared batches. Samples were crushed with a mortar and pestle under an argon atmosphere and immobilised in an eicosane matrix within 400 MHz Wilmad borosilicate NMR tubes to prevent sample reorientation during measurements. The tube was flame-sealed under dynamic vacuum ($1 \times 10^{-3}$ mbar) to a length of approximately 3 cm and mounted in the centre of a drinking straw, with the straw fixed to the end of an MPMS 3 sample rod. Care was taken to ensure complete thermalisation of the sample before each data point was measured by employing delays at each temperature point as well as a slow cooling rate (5 K/min from 300 to 100 K; 1 K/min from 100 to 1.8 K). The sample was held at 1.8 K for 30 min before isothermal magnetisation measurements to account for slow thermal equilibration of the sample. Diamagnetic corrections were applied using tabulated Pascal constants[85]. Measurements were corrected for the effect of the blank sample holders (flame sealed Wilmad NMR tube and straw) and eicosane matrix.

### Modified synthesis of [HNEt₃][BPh₄]

To a vigorously stirring solution of sodium tetraphenylborate (5.13 g, 15.00 mmol) in $H_2O$ (50 mL) was added dropwise a solution of triethylamine hydrochloride (2.47 g, 18.00 mmol) in $H_2O$ (15 mL), and the reaction stirred for 16 h. The resultant white suspension was filtered and the residue washed with $H_2O$ (3 × 10 mL). Removal of volatiles *in vacuo* and further drying at 60 °C afforded $[HNEt_3][BPh_4]$ as a free-flowing white powder, which was used without further purification. Yield: 6.89 g, 90%. $^1H$ ($D_6$-DMSO, 298 K): δ 8.80 (s, 1H, *H*NEt₃), 7.13 (br s, 8H, *meta*-Ar*H*), 6.88 (t, $^3J_{HH}$ = 7.3 Hz, 8H, *ortho*-Ar*H*), 6.75 (m, 4H, *para*-Ar*H*), 3.05 (m, 6H, HN(C*H₂*CH₃)₃), 1.12 (t, $^3J_{HH}$ = 7.3 Hz, 9H, HN(CH₂C*H₃*)₃ ppm. $^{13}C\{^1H\}$ ($D_6$-DMSO, 298 K): δ 136.05 (Ar*C*), 125.89 (Ar*C*), 122.06 (Ar*C*), 46.25 (HN(*C*H₂CH₃)₃), 9.18 (HN(CH₂*C*H₃)₃) ppm. $^{11}B\{^1H\}$ ($D_6$-DMSO, 298 K): δ 7.81 ppm. ATR-IR ν/cm⁻¹: 3128 (m), 3051 (m), 3000 (w), 2986 (w), 1580 (w), 1477 (m), 1426 (m), 1385 (m), 1292 (w), 1268 (w), 1179 (w), 1148 (m), 1064 (m), 1031 (m), 1013 (m), 863 (w), 838 (m), 807 (w), 772 (m), 739 (s), 702 (s), 604 (s), 476 (m), 462 (m), 415 (w) cm⁻¹.

### Modified synthesis of KC₈

A 250 mL round-bottomed Schlenk flask was charged with reagent grade (>99.9%) graphite (3.55 g, 818.4 mmol) and dried under dynamic vacuum ($1 \times 10^{-3}$ mbar) at 100 °C for four hours. In an argon-filled glovebox, freshly cut potassium metal (1.45 g, 102.3 mmol) is added. The mixture is then heated under an argon atmosphere with a blowtorch whilst agitating causing the potassium metal to melt and intercalation to occur. Continue heating until the mixture has completely changed colour from black to bronze, which will be for approximately 3 h. Once the reaction is complete, allow to cool to room temperature. Yield: 5.0 g, 99%.

### Modified synthesis of UCl₄

A 1000 mL round-bottomed flask was charged with $UO_3$ (23.54 g, 82.22 mmol) and hexachloropropene (250 mL). The flask was equipped with two condensers stacked on top of one another, and the flask placed under an inert gas supply. The mixture was heated carefully to reflux, which was accompanied by a violent exotherm and the liberation of a dark brown gas. The flask was lifted away from the heating mantle to allow the exotherm to subside before heating was resumed. **Note:** this moderation of the exotherm step may be needed to be conducted multiple times. The reaction mixture was then left to gently reflux for 16 h. During which time, $UCl_4$ precipitates from solution as a

green solid. The mixture was cooled to room temperature, and the reaction mixture carefully filtered away from the green solid before washing with dichloromethane (3 × 150 mL). Removal of volatiles *in vacuo* afforded $UCl_4$ as a free-flowing green powder, which was used without further purification. Yield: 28.01 g, 90%.

## Modified synthesis of $KCH_2Ph$

To a stirring suspension of KOtBu (5.60 g, 50.00 mmol) in toluene (100 mL) was added nBuLi (20 mL, 2.5 M in hexane, 50.00 mmol) dropwise at 0 °C. The mixture was warmed to room temperature and stirred for an additional 30 min. The orange/red suspension was filtered, and the resultant red solid washed with toluene (2 × 50 mL) and hexane (20 mL). Removal of the volatiles *in vacuo* afforded $KCH_2Ph$ as a free-flowing orange powder, which was used without further purification. Yield: 6.40 g, 98%.

## Modified synthesis of [M(3n-crown-n)$_x$SbH$_2$] (1Na: M = Na, n = 5; 1 K: M = K, n = 6)

**Representative procedure.** Freshly prepared $SbH_3$ starting from 20 g $SbCl_3$ and 5 g $LiAlH_4$[86] was condensed into a solution of MOtBu (M = Na, K) in THF at –78 °C and stirred for 3 h resulting in the formation of a deep red solution. The resultant suspension was then filtered at –72 °C yielding a clear dark red solution. Through the addition of an appropriate crown-ether, **1 M** precipitates from solution and can be separated via filtration at –72 °C before washing with either toluene or pentane. The product is then used without further purification. Apparatus used for the synthesis of **1M** (where M = Na or K) are shown in Supplementary Figs. 10 and 11. **1Na:** 15C5 (2.64 g, 12.0 mmol, 1 eq.), NaOtBu (1.15 g, 12.0 mmol, 1 eq.). Yield: 3.34 g, 76%. $^1$H NMR (D$_5$-pyridine, 298 K): δ 3.60 (s, 20H, 15C5), –3.82 (s, 2H, SbH$_2$) ppm. $^{13}$C{$^1$H} NMR (D$_5$-pyridine, 298 K): δ 69.5 (15C5) ppm. ATR-IR v/cm$^{-1}$: 2943 (w), 2912 (w), 2886 (w), 2865 (w), 1784 (*SbH*) (m), 1611 (m), 1472 (m), 1457 (m), 1347 (m), 1295 (m), 1274 (m), 1260 (m), 1245 (m), 1092 (s), 1044 (s), 1034 (s), 941 (s), 923 (s), 860 (m), 837 (s), 826 (s), 572 (w), 540 (w), 522 (w). **1K:** 18C6 (14.85 g, 56.2 mmol, 1 eq.), KOtBu (6.3 g, 56.2 mmol, 1 eq.). Yield: 18.80 g, 78%. $^1$H NMR (D$_5$-pyridine, 298 K): δ 3.47 (s, 24H, 18C6), –3.73 (s, 2H, SbH$_2$) ppm. $^{13}$C{$^1$H} NMR (D$_5$-pyridine, 298 K): δ 70.9 (18C6) ppm. ATR-IR v/cm$^{-1}$: 2906 (m), 2879 (m), 2819 (m), 1776 (*SbH*) (m), 1467 (m), 1467 (w), 1449 (w), 1432 (w), 1347 (m), 1283 (w), 1245 (m), 1100 (s), 1059 (m), 960 (s), 838 (s), 802 (m), 527 (w).

## Modified synthesis of LiCy

**Under an Ar atmosphere.** To a suspension of lithium granules (0.6% Na, 2.50 g, 360 mmol, 2.2 eq.) in toluene (500 mL) was added cyclohexyl chloride (19.33 mL, 163 mmol) dropwise at room temperature. After the addition was complete, the reaction was heated to 50 °C and stirred for 24 h, during which time there was a colour change to purple. The resultant reaction mixture was allowed to cool to room temperature and filtered to obtain a yellow solution. Removal of the volatiles *in vacuo* yielded LiCy as an orange solid, which was used immediately without further purification. Yield: 9.00 g, 61%.

## Modified synthesis of $Cy_3SiCl$

**Under an Ar atmosphere.** To a slurry of LiCy (9.00 g, 100.00 mmol) in toluene (200 mL) was added $Cy_2SiCl_2$ (26.53 g, 100.00 mmol) dropwise at –78 °C. After the addition was complete, the pale-orange slurry was warmed to room temperature and stirred for a further 24 h, before the resultant pale-orange slurry was filtered to remove LiCl. Removal of the volatiles *in vacuo* yielded $Cy_3SiCl$ as pale-orange solid, which was used without further purification. Yield: 28.80 g, 92%. $^1$H NMR (C$_6$D$_6$, 298 K): δ 1.87–1.83 (m, 6H, Cy*H*), 1.75–1.62 (m, 9H, Cy*H*), 1.40–1.31 (m, 6H, Cy*H*), 1.21–1.13 (m, 8H, Cy*H*), 1.10–1.01 (m, 4H, Cy*H*) ppm. $^{29}$Si NMR (C$_6$D$_6$, 298 K): δ 27.48 ppm.

## Modified synthesis of $Tren^{TIPS}Li_3$

$N(CH_2CH_2NH_2)_3$ (10 mL, 66.60 mmol) was dissolved in THF (50 mL). nBuLi (2.5 M, 80 mL, 200.00 mmol) was added dropwise at –78 °C, warmed to room temperature and the mixture stirred for 6 h. The solution was then cooled to –78 °C, and ClSi$^i$Pr$_3$ (42.80 mL, 200.00 mmol) was added in a portion-wise manner and the solution stirred at room temperature for 16 h. Removal of volatiles *in vacuo* resulted in a pale-yellow sticky solid. The product was extracted with hexane (2 × 50 mL), and the solution was filtered from the LiCl precipitate. nBuLi (2.5 M, 80 mL, 200.00 mmol) was added dropwise at –78 °C, warmed to room temperature and the solution was stirred for 6 h at room temperature. Removal of volatiles *in vacuo* resulted in an off-white solid which was washed with cold hexane (2 × 10 mL) to yield $Tren^{TIPS}Li_3$ as a white powder. Colourless crystals of $Tren^{TIPS}Li_3$ were grown from a concentrated solution in hexane stored at –30 °C. Yield: 34.25 g, 81%. $^1$H NMR (C$_6$D$_6$, 298 K): δ 3.22 (t, 6H, NCH$_2$C*H$_2$*), 2.38 (t, 6H, NC*H$_2$*CH$_2$), 1.29 (m, 9H, C*H*(CH$_3$)$_2$), 1.28 (m, 54H, CH(C*H$_3$*)$_2$) ppm. ATR-IR v/cm$^{-1}$: 2936 (s), 2883 (m), 2854 (s), 2770 (w), 2661 (w), 1461 (s), 1362 (w), 1342 (w), 1270 (w), 1237 (w), 1138 (w), 1056 (s), 1027 (s), 1004 (w), 992 (w), 933 (s), 877 (s), 776 (s), 657 (s), 637 (s), 567 (m), 513 (w), 493 (s), 462 (w), 448 (w), 419 (m) cm$^{-1}$.

## Modified synthesis of $Tren^{TCHS}Li_3$

$N(CH_2CH_2NH_2)_3$ (2.93 g, 20.00 mmol) was dissolved in THF (50 mL). nBuLi (2.5 M, 24 ml, 60.00 mmol) was added dropwise at –78 °C, warmed to room temperature and the mixture was stirred for 6 h. The solution was then cooled to –78 °C, and $Cy_3SiCl$ (18.78 g, 60.00 mmol) was added in a portion-wise manner and the solution was stirred at room temperature for 16 h. Removal of volatiles *in vacuo* resulted in a pale-yellow sticky solid. The product was extracted with hexane (2 × 50 mL), and the solution was filtered from the LiCl precipitate. nBuLi (2.5 M, 24 mL, 60.00 mmol) was added dropwise at –78 °C, warmed to room temperature and the solution was stirred for 6 h at room temperature. Removal of volatiles *in vacuo* resulted in an off-white solid which was washed with pentane (2 × 10 mL) to yield $Tren^{TCHS}Li_3$ as an off-white powder. Colourless crystals of $Tren^{TIPS}Li_3$ were grown from a concentrated solution in benzene at room temperature. Yield: 11.12 g, 56%. $^1$H NMR (C$_6$D$_6$, 298 K): δ 3.18 (t, $^3J_{HH}$ = 5.3 Hz, 6H, NCH$_2$C*H$_2$*), 2.35 (t, $^3J_{HH}$ = 5.3 Hz, 6H, NC*H$_2$*CH$_2$), 1.93–1.88 (m, 45H, Cy*H*), 1.50–1.38 (m, 54H, Cy*H*) ppm. ATR-IR v/cm$^{-1}$: 2912 (s), 2842 (s), 2649 (w), 1442 (s), 1339 (w), 1261 (w), 1185 (w), 1163 (w), 1099 (w), 1066 (m), 1029 (m), 994 (m), 941 (s), 885 (m), 846 (m), 813 (m), 768 (s), 729 (m), 711 (w), 678 (w), 589 (w), 524 (s), 468 (s), 425 (w) cm$^{-1}$.

## Modified synthesis of [U(Tren$^{TIPS}$)Cl]

A solution of $Tren^{TIPS}Li_3$ (12.66 g, 20.00 mmol) in THF (50 mL) was added dropwise to a stirring solution of $UCl_4$ (7.60 g, 20.00 mmol) in THF (80 mL) at –78 °C. The mixture was allowed to warm to room temperature before stirring for 16 h. Removal of volatiles *in vacuo* resulted in a brown solid. The product was extracted in hot toluene (100 mL) and the solution was filtered from the LiCl precipitate. Removal of volatiles *in vacuo* resulted in a pale-brown solid. The product was washed with hexane (2 × 10 mL) to yield [U(Tren$^{TIPS}$)Cl] as a brown solid. Green crystals of [U(Tren$^{TIPS}$)Cl] were grown from a concentrated solution in toluene at room temperature. Yield: 10.75 g, 61%. $^1$H NMR (C$_6$D$_6$, 298 K): δ 9.20 (s, 54H, CH(C*H$_3$*)$_2$), 8.42 (s, 9H, C*H*(CH$_3$)$_2$), 5.72 (s, 6H, NCH$_2$C*H$_2$*), –36.20 (s, 6H, NC*H$_2$*CH$_2$) ppm. ATR-IR v/cm$^{-1}$: 2938 (m), 2922 (m), 2860 (m), 1463 (s), 1383 (w), 1360 (w), 1339 (w), 1272 (w), 1132 (w), 1039 (m), 1009 (m), 988 (w), 918 (s), 879 (s), 817 (w), 723 (s), 672 (s), 626 (s), 548 (m), 515 (m), 448 (m).

## Modified synthesis of [U(Tren$^{TCHS}$)Cl]

A suspension of $Tren^{TCHS}Li_3$ (5.00 g, 5.00 mmol) in THF (50 mL) was added in a dropwise manner to a stirring solution of $UCl_4$ (1.90 g,

5.00 mmol) in THF (50 mL) at −78 °C. The mixture was allowed to warm to room temperature before stirring for 16 h, during which time, the product precipitated out as a green solid. The reaction mixture was concentrated to nearly half-volume (50 mL) and the mixture was filtered to remove the LiCl by-product dissolved in the dark brown solution phase in THF. The product was further washed with $Et_2O$ (2 × 20 mL) followed by hexane (20 mL) and then dried *in vacuo* to yield [U(Tren$^{TCHS}$)Cl] as a green solid. Green crystals of [U(Tren$^{TCHS}$)Cl] were grown through the cooling of a hot (100 °C) concentrated solution in toluene. Yield: 3.13 g, 51%. $^1H$ NMR ($C_6D_6$, 298 K): δ 12.09 (br s, 18H, Cy*H*), 11.78 (br s, 17H, Cy*H*), 8.85 (br s, 7H, Cy*H*), 6.13 (br s, 6H, NCH$_2$C*H$_2$*), 3.44 (br s, 7H, Cy*H*), 3.36 (br s, 20H, Cy*H*), 2.26 (br s, 8H, Cy*H*), 1.29 (br s, 8H, Cy*H*), −0.04 (br, 14H, Cy*H*), −39.15 (br s, 6H, NC*H$_2$*CH$_2$) ppm. ATR-IR v/cm$^{-1}$: 2914 (s), 2844 (s), 2166 (w), 1444 (s), 1346 (w), 1325 (w), 1288 (w), 1274 (w), 1204 (w), 1167 (w), 1136 (w), 1105 (m), 1070 (m), 1048 (m), 998 (m), 931 (m), 908 (m), 891 (s), 842 (m), 813 (m), 746 (s), 725 (s), 684 (m), 587 (m), 538 (s), 519 (m), 485 (m), 437 (m) cm$^{-1}$.

### Modified synthesis of [U(Tren$^{TIPS}$)]

A suspension of [U(Tren$^{TIPS}$)Cl] (4.42 g, 5.00 mmol) in hexane (20 mL) was transferred into a Schlenk flask containing a freshly prepared potassium mirror (20-fold excess) and a glass-coated stirrer bar. The mixture was stirred vigorously for seven days, after which the resultant deep purple solution was filtered into another Schlenk flask containing a freshly prepared potassium mirror (20-fold excess) and a glass-coated stirrer bar. The mixture was stirred vigorously for a further five days. Filtration of the deep purple solution and removal of volatiles *in vacuo* yielded [U(Tren$^{TIPS}$)] as a dark purple powder. Dark purple crystals of [U(Tren$^{TIPS}$)] were grown through the storage of a concentrated hexane solution at −30 °C. Yield: 1.33 g, 78%. $^1H$ NMR ($C_6D_6$, 298 K): δ 17.67 (s, 6H, C*H$_2$*), 7.22 (s, 9H, C*H*(CH$_3$)$_2$), 4.03 (s, 54H, CH(C*H$_3$*)$_2$), −40.07 (s, 6H, C*H$_2$*) ppm. ATR-IR v/cm$^{-1}$: 2936 (s), 2885 (m), 2858 (s), 2751 (m), 2661 (w), 1459 (w), 1385 (w), 1362 (w), 1344 (w), 1257 (m), 1134 (w), 1093 (s), 1070 (s), 1035 (w), 1000 (m), 926 (m), 912 (s), 877 (s), 772 (s), 750 (s), 735 (s), 663 (s), 628 (s), 589 (m), 567 (m), 544 (m), 509 (m), 497 (m), 417 (m) cm$^{-1}$.

### Modified synthesis of 2U

A Schlenk flask was charged with [U(Tren$^{TIPS}$)Cl] (9.00 g, 10.00 mmol) and KCH$_2$Ph (1.30 g, 10.00 mmol). At −78 °C, toluene (100 mL) was added with stirring before the mixture was warmed to room temperature and stirred for a further 16 h to afford an orange-brown mixture. Removal of volatiles *in vacuo* resulted in a sticky red solid. The product was extracted in hexane (3 ×50 mL), filtered and volatiles removed *in vacuo* before being washed with cold pentane (2 × 10 mL) to yield 2U as a red solid, which was further dried *in vacuo*. Orange crystals of 2U were grown through the storage of a concentrated pentane solution at 4 °C for 72 h. Yield: 0.55 g, 65%. $^1H$ NMR ($C_6D_6$, 298 K): δ 46.75 (1H, s, C*H$_2$*), 32.65 (1H, s, C*H$_2$*), 28.89 (1H, s, C*H$_2$*), 23.20 (1H, s, C*H$_2$*), 15.93 (1H, s, UCH$_2$C*H*), 15.76 (3H, s, UCH$_2$CH*Me*), 11.73 (1H, s, C*H$_2$*), 11.30 (1H, m, C*H*Me$_2$), 9.54 (1H, m, C*H*Me$_2$), 8.38 (3H, d, $^3J_{HH}$ = 7.3 Hz, Si*Me*), 6.82 (1H, s, C*H$_2$*), 6.02 (3H, d, $^3J_{HH}$ = 7.3 Hz, Si*Me*), 5.65 (3H, d, $^3J_{HH}$ = 6.4 Hz, Si*Me*), 4.01 (9H, s, 3×Si*Me*), 3.72 (3H, d, $^3J_{HH}$ = 6.4 Hz, Si*Me*), 2.77 (1H, s, C*H$_2$*), 1.52 (3H, s, 3×C*H*Me$_2$), − 0.53 (1H, s, C*H$_2$*), −1.33 (2H, s, UC*H$_2$*), −2.61 (9H, 3×Si*Me*), −2.89 (9H, s, 3×Si*Me*), −4.70 (9H, s, 3×Si*Me*), −20.84 (3H, s, 3×C*H*Me$_2$), −23.09 (1H, s, C*H$_2$*), −24.14 (1H, s, C*H$_2$*), −31.64 (1H, s, C*H$_2$*), −35.20 (1H, s, C*H$_2$*) ppm. ATR-IR v/cm$^{-1}$: 2938 (s), 2885 (m), 2858 (s), 2778 (m), 2721 (w), 2665 (w), 1236 (s), 1379 (w), 1362 (w), 1346 (w), 1261 (m), 1136 (w), 1089 (m), 1068 (s), 1009 (m), 906 (m), 879 (s), 762 (s), 737 (s), 669 (s), 628 (s), 587 (s), 550 (m), 513 (m), 474 (w), 406 (m) cm$^{-1}$.

### Modified synthesis of 3U

A Schlenk flask was charged with [U(Tren$^{TCHS}$)Cl] (2.50 g, 2.00 mmol) and Me$_3$SiCH$_2$Li (2.10 g, 2.20 mmol). At −78 °C, toluene (100 mL) was added with stirring before the mixture was warmed to room

temperature and stirred for a further 24 h to afford a green slurry. The reaction was then heated to 100 °C and filtered from the LiCl by-product. Removal of the volatiles *in vacuo* followed by washing with pentane (2 × 30 mL) to yield 3U as red solid, which was further dried *in vacuo*. Red crystals of 3U were grown through the cooling of a hot (100 °C) concentrated toluene solution. Yield: 1.65 g, 68%. The insolubility of 3U precluded the acquisition of NMR spectroscopic data. ATR-IR v/cm$^{-1}$: 2910 (s), 2842 (s), 2657 (w), 1444 (s), 1342 (w), 1259 (s), 1196 (w), 1183 (w), 1169 (w), 1093 (m), 1070 (m), 1037 (m), 1017 (m), 996 (m), 918 (m), 889 (m), 840 (m), 815 (m), 741 (s), 721 (s), 678 (m), 550 (m), 522 (s), 480 (w), 448 (w), 435 (w) cm$^{-1}$.

### Modified synthesis of 4U

To a THF (50 mL) solution of 2U was added [HNEt$_3$][BPh$_4$] (3.33 g, 7.9 mmol) in a portion-wise manner at −78 °C. The resultant orange slurry was warmed to room temperature and stirred for a further 16 h to afford a yellow-green solution. Removal of the volatiles *in vacuo* afforded a yellow-green oil to which hexane (50 ml) was added. The mixture was heated briefly to 60 °C and allowed to cool slowly to room temperature whilst stirring vigorously. Trituration was complete inside 2 h affording a pale green solid which was isolated by filtration, washed with hexane (3 × 5 mL) and dried *in vacuo*. Green crystals of 4U were grown through the storage of a concentrated toluene solution at −30 °C. Yield: 9.02 g, 92%. $^1H$ NMR ($C_6D_6$, 298 K): δ 28.32 (s, 6H, C*H$_2$*), 7.03 (s, 4H, Ar-H), 5.80 (s, 9H, C*H*(CH$_3$)$_2$), 4.08 (s, 4H, THF), 3.06 (s, 54H, CH(C*H$_3$*)$_2$), 2.46 (s, 4H, THF), 2.12 (s, 8H, Ar-*H*), 0.82 (s, 8H, Ar-*H*), −49.51 (s, 6H, C*H$_2$*) ppm. ATR-IR v/cm$^{-1}$: 3058 (w), 3035 (w), 2938 (m), 2862 (m), 2731 (m), 1580 (w), 1463 (m), 1424 (w), 1387 (w), 1366 (w), 1288 (w), 1257 (m), 1128 (m), 1064 (s), 1031 (m), 1007 (m), 924 (m), 891 (s), 842 (w), 762 (s), 733 (s), 704 (s), 672 (s), 639 (s), 612 (s), 585 (s), 511 (w), 470 (m), 419 (m) cm$^{-1}$.

### Synthesis and isolation of [Na(15C5)(OEt$_2$)][{U(Tren$^{TIPS}$)}$_2$(μ-Sb)] (5UNa)

In the strict absence of light, 1Na (0.073 g, 0.20 mmol) was added in a portion-wise manner to a red stirring solution of 2U (0.34 g, 0.40 mmol) in benzene (10 mL) at room temperature. The resultant brown solution was stirred overnight, before Celite® was added into the reaction with stirring and then the mixture filtered to yield a dark brown solution. Removal of the volatiles *in vacuo* yielded a brown solid which was dried for one hour. Crystals of 5UNa suitable for single-crystal X-ray diffraction studies were grown through the slow evaporation of a concentrated $Et_2O$ (10 mL) solution at room temperature; in bulk these crystals appear to be black, but they are actually dark red-black when inspected in detail. The mother liquor was decanted, and the black crystals were washed with $Et_2O$ (2 × 2 mL) and dried *in vacuo* to afford 5UNa. Yield: 0.051 g, 15%. Anal. Calcd for C$_{82}$H$_{185}$N$_8$NaO$_{6.5}$SbSi$_6$U$_2$: C, 45.25; H, 8.57; N, 5.15%; Found: C, 45.25; H, 8.74; N, 4.98%. $^1H$ NMR (D$_8$-THF, 298 K): δ 21.06 (br, 12H, NC*H$_2$*CH$_2$), 4.54 (br, 12H, NCH$_2$C*H$_2$*), 3.38 (m, 6H, (C*H$_3$*CH$_2$)$_2$O), 1.12 (m, 4H, (CH$_3$C*H$_2$*)$_2$O), − 0.07 (br, 18H, C*H*(CH$_3$)$_2$), − 2.40 (br, 108H, CH(C*H$_3$*)$_2$) ppm. The resonances attributed to the 20 hydrogens of 15C5 could not be definitively assigned due to overlapping with resonances from the D$_8$-THF solvent used for the NMR experiments. Despite repeated attempts, no signal could be observed in the $^{29}$Si{$^1$H} NMR spectrum of 5UNa. ATR-IR v/cm$^{-1}$: 2932 (m), 2856 (s), 1459 (m), 1383 (w), 1352 (w), 1333 (w), 1290 (w), 1288 (w), 1261 (w), 1243 (s), 1115 (s), 1093 (m), 1041 (s), 1013 (w), 993 (m), 881 (s), 787 (vs), 735 (s), 669 (s), 567 (w). UV-vis-NIR (THF): λ$_{max}$ (v/cm$^{-1}$; ε/M$^{-1}$cm$^{-1}$) = 1,208 (8,275; 607), 832 (12,020; 1,506), 742 (13,477; 1,469), 575 (17,378; 2,278), 462 (21,640; 6,418) nm. Sample decomposition precluded the acquisition of Raman spectroscopic data.

### Synthesis and isolation of [K(18C6)][{U(Tren$^{TIPS}$)}$_2$(μ-Sb)] (5UK)

In the strict absence of light, 1 K (0.25 g, 0.50 mmol) was added in a portion-wise manner to a red stirring solution of 2U (0.85 g, 1.00 mmol)

in benzene (10 mL) at room temperature. The resultant brown solution was stirred for two hours, before Celite® was added into the reaction with stirring and then the mixture filtered to yield a dark brown solution. Removal of the volatiles *in vacuo* yielded a brown solid which was dried for one hour. Crystals suitable for single-crystal X-ray diffraction studies were grown through the slow evaporation of a concentrated $Et_2O$ (10 mL) solution at room temperature; in bulk these crystals appear to be black, but they are actually dark red-black when inspected in detail. The mother liquor was decanted, and the black crystals were further washed with $Et_2O$ (2 × 2 mL) and dried *in vacuo* for 30 min to afford **5UK**. Yield: 0.51 g, 46%. Anal. Calcd for $C_{78}H_{174}KN_8O_6SbSi_6U_2$: C, 44.07; H, 8.25; N, 5.27%; Found: C, 44.42; H, 8.30; N, 5.17%. $^1H$ NMR ($D_8$-THF, 298 K): δ 21.46 (br, 12H, $NCH_2CH_2$), 4.61 (br, 12H, $NCH_2CH_2$), 3.44 (s, 24H, $[C_2H_4O]_6$), −0.08 (br, 18H, $CH(CH_3)_2$), −2.44 (br, 108H, $CH(CH_3)_2$) ppm. $^{29}Si\{^1H\}$ NMR ($D_8$-THF, 298 K): δ −70.77 ppm. ATR-IR v/cm$^{-1}$: 2911 (m), 2855 (s), 1456 (m), 1381 (w), 1352 (w), 1335 (w), 1274 (w), 1249 (w), 1105 (s), 1043 (s), 1016 (m), 962 (m), 933 (s), 880 (s), 786 (m), 735 (vs), 671 (s), 625 (m), 567 (w), 515 (w), 443 (w). UV-vis-NIR (THF): $\lambda_{max}$ (v/cm$^{-1}$; ε/M$^{-1}$cm$^{-1}$) = 1,211 (8,257; 678), 1,112 (8,996; 731), 834 (11,996; 1,784), 746 (13,400; 1,664), 572 (17,471; 2,571), 519 (19,271; 4,226), 498 (20,086; 3,980) nm. Sample decomposition precluded the acquisition of Raman spectroscopic data.

## Synthesis and isolation of [K(2.2.2-cryptand)] [{U(Tren$^{TIPS}$)}$_2$(μ-Sb)] (5UK′)

*Method A*: A 20 mL glass scintillation vial was charged with a PTFE-coated stirrer bar and (**6U/7U**, see later for details) (0.19 g, 0.05 mmol), $KC_8$ (0.02 g, 0.15 mmol), and 2.2.2-cryptand (0.04 g, 0.11 mmol). At room temperature and in the strict absence of light, benzene (10 mL) was added resulting in the formation of a dark brown suspension, which was stirred for 24 h. Removal of the volatiles *in vacuo* and extraction of soluble residues with THF (5 mL) yielded a brown suspension. Removal of the volatiles *in vacuo* yielded a black oily solid which was washed with $Et_2O$ (2 × 2 mL) and dried *in vacuo* for one hour, affording **5UK′** as a black solid. Crystals of **5UK′** suitable for single crystal X-ray diffraction studies were obtained by the diffusion of pentane into a concentrated THF solution at room temperature; in bulk these crystals appear to be black, but they are actually dark red-black when inspected in detail. Yield: 0.056 g, 24% (by U content). *Method B*: A 20 mL glass scintillation vial was charged with a PTFE-coated stirrer bar and solid **5UK** (0.43 g, 0.20 mmol). At room temperature, THF (10 mL) was added resulting in the formation of a dark brown solution. A colourless solution of 2.2.2-cryptand (0.08 g, 0.20 mmol) in THF (2 mL) was then added in a portion-wise manner with stirring resulting in the immediate formation of a dark brown solution which was stirred for 24 h. Removal of the volatiles *in vacuo* yielded a black oily solid which was washed with $Et_2O$ (2 × 2 mL) and dried *in vacuo* for one hour to afford **5UK′**. Yield: 0.32 g, 72%. Anal. Calcd for $C_{88}H_{196}KN_{10}O_7SbSi_6U_2$: C, 45.72; H, 8.55; N, 6.06%; Found: C, 46.08; H, 8.75; N, 5.73%. $^1H$ NMR ($D_8$-THF, 298 K): δ 20.93 (br, 12H, $NCH_2CH_2$), 4.54 (br, 12H, $NCH_2CH_2$), 3.36 (s, 12H, $OCH_2CH_2O$), 3.32 (t, 12H, $NCH_2CH_2O$), 2.33 (t, 12H, $NCH_2CH_2O$), −0.04 (br, 18H, $CH(CH_3)_2$), −2.36 (br, 108H, $CH(CH_3)_2$) ppm. $^{29}Si\{^1H\}$ NMR ($D_8$-THF, 298 K): δ −67.98 ppm. ATR-IR v/cm$^{-1}$: 2934 (w), 2880 (m), 2854 (s), 1458 (m), 1445 (m), 1381 (w), 1356 (w), 1335 (w), 1296 (w), 1259 (w), 1133 (s), 1104 (m), 1044 (s), 1016 (m), 931 (s), 880 (s), 784 (s), 732 (vs), 670 (s), 629 (m), 567 (w), 515 (w), 436 (w). UV-vis-NIR (THF): $\lambda_{max}$ (v/cm$^{-1}$; ε/M$^{-1}$cm$^{-1}$) = 1,211 (8,257; 807), 1,109 (9,019; 874), 839 (11,914; 2,129), 748 (13,371; 1,983), 580 (17,229; 2,921) nm. Sample decomposition precluded the acquisition of Raman spectroscopic data.

## Synthesis and isolation of [{U(Tren$^{TIPS}$)}$_2$(μ-SbH)] (6U)

*Method A*: In the strict absence of light, **1Na** (0.073 g, 0.20 mmol) was added in a portion-wise manner to a yellow stirring slurry of **4U** (0.25 g,

0.20 mmol) in benzene (10 mL). The resultant red brown suspension was stirred overnight, before Celite® was added into the reaction with stirring. The mixture was then heated up to 80 °C and filtered to yield a dark brown solution. Removal of the volatiles *in vacuo* yielded a green/brown oily solid which was washed with $Et_2O$ (2 × 5 mL) and dried *in vacuo* for one hour, affording **6U** as green/brown solid. Crystals suitable for single-crystal X-ray diffraction studies were grown by heating a benzene solution to 80 °C and slowly cooling to room temperature; in bulk these crystals appear to be black, but they are actually dark red-black when inspected in detail. Yield: 0.11 g, 30%. *Method B*: A 20 mL glass scintillation vial was charged with a PTFE-coated stirrer bar and solid **5UK** (0.43 g, 0.20 mmol). At room temperature, THF (5 mL) was added resulting in the formation of a dark brown solution. A colourless solution of [HNEt$_3$][BPh$_4$] (0.09 g, 0.20 mmol) in THF (2 mL) was then added with stirring resulting in the immediate formation of a brown suspension which was stirred overnight. Celite® added into the reaction with stirring and then the mixture filtered to yield a dark brown solution. Removal of the volatiles *in vacuo* yielded a dark brown solid which was dried for two hours. Black crystals suitable for single-crystal X-ray diffraction studies were grown through the slow evaporation of a concentrated $Et_2O$ (10 mL) solution. The mother liquor was decanted, and the black crystals were washed with $Et_2O$ (2 × 2 mL) and dried *in vacuo* to afford **6U**. Yield: 0.12 g, 31%. *Method C*: The same procedure as described in *Method B* but using **5UNa** (0.38 g, 0.20 mmol) and [HNEt$_3$][BPh$_4$] (0.09 g, 0.20 mmol) in THF (5 mL). Yield: 0.09 g, 25%. Anal. Calcd for $C_{66}H_{151}N_8SbSi_6U_2$: C, 43.48; H, 8.35; N, 6.15%; Found: C, 43.26; H, 8.42; N, 6.08%. ATR-IR v/cm$^{-1}$: 2941 (m), 2887 (m), 2860 (m), 2842 (m), 1459 (m), 1381 (w), 1362 (w), 1336 (w), 1271 (w), 1253 (w), 1140 (m), 1040 (m), 1035 (m), 1010 (m), 929 (s), 880 (s), 795 (m), 720 (vs), 673 (m), 628 (m), 569 (w), 517 (w), 442 (w). Complex **6U** is insoluble in common arene and ethereal solvents after being isolated as a crystalline solid which prohibited the acquisition of solution-state NMR and UV/Vis/NIR spectroscopic data. Sample decomposition precluded the acquisition of Raman spectroscopic data.

## Synthesis and isolation of [{U(Tren$^{TIPS}$)}$_2$(μ-η$^2$:η$^2$-Sb$_2$)] (7U) with 6U

In the strict absence of light, **1K** (0.20 g, 0.40 mmol) was added in a portion-wise manner to a yellow stirring slurry of **4U** (0.50 g, 0.40 mmol) in benzene (10 mL). The resultant red brown suspension was stirred overnight, before Celite® was added into the reaction with stirring. The mixture was then heated up to 80 °C and filtered to yield a dark brown solution. Removal of the volatiles *in vacuo* yielded a green/brown oily solid which was washed with $Et_2O$ (2 × 5 mL) and dried *in vacuo* for one hour, affording **7U** as a green/brown solid. Crystals suitable for single-crystal X-ray diffraction studies were grown by heating a benzene solution to 80 °C and cooling slowly to room temperature; in bulk these crystals appear to be black, but they are actually dark red-black when inspected in detail. The mother liquor was decanted, and the black crystals were washed with benzene (2 × 2 mL) and dried *in vacuo* to afford **7U**. Yield: 0.11 g, 26%. Anal. Calcd for $C_{66}H_{150}N_8Sb_2Si_6U_2$(Et$_2$O): C, 41.66; H, 7.99; N, 5.55%; Found: C, 41.74; H, 7.87; N, 5.43%. $^1H$ NMR ($C_6D_6$, 298 K): δ 8.64 (br, 108H, $CH(CH_3)_2$), 4.43 (br, 18H, $CH(CH_3)_2$), −6.13 (br, 12H, $NCH_2CH_2$), −34.13 (br, 12H, $NCH_2CH_2$) ppm. $^{29}Si\{^1H\}$ NMR ($C_6D_6$, 298 K): δ 25.87 ppm. ATR-IR v/cm$^{-1}$: 2939 (m), 2861 (s), 1462 (m), 1385 (w), 1271 (w), 1253 (w), 1106 (w), 1064 (w), 1032 (m), 1010 (m), 930 (7 s), 882 (s), 800 (m), 765 (vs), 730 (s), 705 (m), 674 (m), 634 (w), 612 (w), 586 (w), 570 (w), 516 (w), 470 (w), 413 (w). Raman v/cm$^{-1}$ (532 nm): 3048 (m), 2922 (w), 2884 (w), 1589 (m), 1046 (w), 1008 (m), 158 (s), 89 (m). UV-vis-NIR (THF): $\lambda_{max}$ (v/cm$^{-1}$; ε/M$^{-1}$cm$^{-1}$) = 1,135 (8,811; 357), 745 (13,417; 1,323), 692 (14,448; 1502), 517 (19,327; 2,534), 472 (21,164; 4,250) nm. However, crystals of **7U** revealed it to be co-crystallised **6U:7U** in the ratio 11:89, which is similar to the thorium analogue[16].

## Synthesis and isolation of [{U(Tren^TIPS)(μ-NK)}₂] (8U)

In the strict absence of light, benzene (10 mL) was added to a mixture of **6U:7U** (0.194 g, 0.1 mmol) and $KC_8$ (0.054 g, 0.4 mmol) at room temperature. The resultant brown solution with black precipitate was stirred for 2 h, before Celite® was added into the reaction with stirring. The mixture was then heated up to 80 °C and filtered to yield a dark brown solution. Dark red crystals of **8U** suitable for single-crystal X-ray diffraction studies were grown by heating a benzene solution to 80 °C and slowly cooling to room temperature. Yield: 0.035 g, 19%. Characterisation data for **8U** align with those of an authentic sample prepared from [U(Tren^TIPS)(N₃)] and $KC_8$[39]. A new single crystal X-ray diffraction structure of **8U** can be found in the Supplementary Information (Supplementary Fig. 6). Here we report IR data from ATR and compare to prior data recorded in a Nujol mull. ATR-IR v/cm⁻¹: 2924 (m), 2850 (m), 1459 (w), 1461 (s), 1399 (w), 1379 (w), 1241 (m), 1109 (m), 1054 (m), 990 (m), 931 (m), 879 (2), 830 (w), 795 (w), 731 (m), 665 (w), 647 (w), 628 (w), 548 (w), 507 (w). FTIR v/cm⁻¹ (Nujol): 1344 (w), 1134 (w), 1069 (s), 1056 (s), 992 (w), 932 (s), 880 (m), 851 (w), 780 (m), 746 (s), 671 (m), 626 (w), 571 (w), 545 (w), 515 (w).

## Reduction of 4U

In the strict absence of light, benzene (10 mL) was added to a solid mixture of **4U** (0.25 g, 0.2 mmol) and $KC_8$ (0.108 g, 0.8 mmol) at room temperature. The resultant yellow/brown suspension was stirred for two hours, before filtering to yield a brown solution. Removal of the volatiles *in vacuo* yielded a sticky brown residue. NMR spectroscopic analysis of the reaction mixture showed the presence of [U(Tren^TIPS)][42] and Tren^TIPS$H_3$[42], as well as unidentifiable, paramagnetically shifted, resonances. ¹H NMR (C₆D₆, 298 K): 29.70 (s), 17.67 (br, s, 6H, NCH₂CH₂, [U(Tren^TIPS)]), 16.06 (br, s), 15.97 (s), 12.06 (s), 11.38 (s), 9.55 (s), 8.42 (s), 7.12 (br, 9H, CH(CH₃)₂, [U(Tren^TIPS)]), 6.01 (s), 5.66 (s), 4.03 (s, 54H, CH(CH₃)₂, [U(Tren^TIPS)]), 3.61 (s), 3.03 (s), 2.89 (s, 6H, NCH₂CH₂, Tren^TIPSH₃), 2.44 (s, 6H, NCH₂CH₂, Tren^TIPSH₃), 1.15 − 0.95 (m, 63H, CH(CH₃)₂ and CH(CH₃)₂, Tren^TIPSH₃), 0.77 (s, 3H, NCH₂CH₂NH, Tren^TIPSH₃), −2.79 (s), −3.14 (s), −4.10 (s), −4.83 (s), −5.28 (s), −8.69 (s), −9.28 (s), −12.36 (s), −21.48 (s), −23.63 (s), −24.71 (s), −40.09 (br, s, 6H, NCH₂CH₂, [U(Tren^TIPS)]) ppm.

## Attempted reduction of [U(Tren^TIPS)]

In the strict absence of light, benzene (10 mL) was added to a solid mixture of [U(Tren^TIPS)] (0.084 g, 0.1 mmol) and $KC_8$ (0.054 g, 0.4 mmol) at room temperature. The resultant purple suspension was stirred for two hours, before filtering to yield a dark purple solution. Removal of the volatiles *in vacuo* yielded a purple/brown solid. NMR spectroscopic analysis of the reaction mixture showed the presence of [U(Tren^TIPS)][42] and Tren^TIPS$H_3$[42], as well as unidentifiable, paramagnetically shifted, resonances. Longer reaction times resulted in no change in the outcome. ¹H NMR (C₆D₆, 298 K): 17.75 (br, s, 6H, NCH₂CH₂, [U(Tren^TIPS)]), 15.78 (s), 8.33 (s), 7.12 (br, 9H, CH(CH₃)₂, [U(Tren^TIPS)]), 5.98 (s), 5.51 (s), 4.03 (s, 54H, CH(CH₃)₂, [U(Tren^TIPS)]), 3.59 (s), 3.03 (s), 2.89 (s, 6H, NCH₂CH₂, Tren^TIPSH₃), 2.44 (s, 6H, NCH₂CH₂, Tren^TIPSH₃), 1.15−0.95 (m, 63H, CH(CH₃)₂ and CH(CH₃)₂, Tren^TIPSH₃), 0.77 (s, 3H, NCH₂CH₂NH, Tren^TIPSH₃), −2.78 (s), −3.09 (s), −4.84 (s), −5.24 (s), −40.09 (br, s, 6H, NCH₂CH₂, [U(Tren^TIPS)]) ppm.

## Reduction of 6U with KC₈

In the strict absence of light, benzene (10 mL) was added to a solid mixture of **6U** (0.026 g, 0.015 mmol) and $KC_8$ (0.007 g, 0.06 mmol) at room temperature. The resultant brown suspension was stirred for two hours, before filtering to yield a brown solution. Removal of the volatiles *in vacuo* yielded a brown solid. NMR spectroscopic analysis of the reaction mixture showed the presence of [U(Tren^TIPS)][42] and Tren^TIPS$H_3$[42], as well as unidentifiable, paramagnetically shifted, resonances. ¹H NMR (C₆D₆, 298 K): 29.81 (s), 17.79 (s, 6H, NCH₂CH₂, [U(Tren^TIPS)]), 12.98 (s), 12.08 (s), 10.50 (s), 9.27 (s), 7.11 (br, 9H,

CH(CH₃)₂, [U(Tren^TIPS)]), 4.03 (s, 54H, CH(CH₃)₂, [U(Tren^TIPS)]), 2.89 (s, 6H, NCH₂CH₂, Tren^TIPSH₃), 2.44 (s, 6H, NCH₂CH₂, Tren^TIPSH₃), 1.15−0.95 (m, 63H, CH(CH₃)₂ and CH(CH₃)₂, Tren^TIPSH₃), 0.77 (s, 3H, NCH₂CH₂NH, Tren^TIPSH₃), −5.29 (s), −8.69 (s), −10.04 (s), −41.09 (s, 6H, NCH₂CH₂, [U(Tren^TIPS)]) ppm.

## Reduction of 6U with KC₈ in the presence of 2.2.2-cryptand

In the strict absence of light, benzene (10 mL) was added to a solid mixture of **6U** (0.017 g, 0.01 mmol), $KC_8$ (0.005 g, 0.04 mmol) and 2.2.2-cryptand (0.015 g, 0.04 mmol) at room temperature. The resultant brown suspension was stirred for two hours, before filtering to yield a brown solution. Removal of the volatiles *in vacuo* yielded a brown oil. NMR spectroscopic analysis of the reaction mixture showed the presence of 2.2.2-cryptand and Tren^TIPS$H_3$[42], as well as unidentifiable, paramagnetically shifted, resonances. ¹H NMR (C₆D₆, 298 K): 28.11 (s), 24.36 (s), 21.89 (s), 21.08 (s), 19.72 (s), 16.75 (s), 15.25 (s), 11.06 (s), 8.67 (s), 8.03 (s), 6.60 (s), 5.46 (s), 3.66 (s, 12H, OCH₂CH₂O), 3.50 (s, 12H, NCH₂CH₂O), 3.33 (s), 2.87 (s, 6H, NCH₂CH₂, Tren^TIPSH₃), 2.53 (s, 12H, NCH₂CH₂O), 2.42 (s, 6H, NCH₂CH₂, Tren^TIPSH₃), 2.33 (s), 1.34 (br), 1.15−0.95 (m, 63H, CH(CH₃)₂ and CH(CH₃)₂, Tren^TIPSH₃), 0.90 (s), 0.77 (s, 3H, NCH₂CH₂NH, Tren^TIPSH₃), 0.19 (s), −3.82 (s), −4.43 (s), −5.56 (s), −8.74 (s), −9.00 (s), −9.61 (s), −11.33 (s), −11.84 (s), −13.34 (s), −14.07 (s), −17.33 (s) ppm.

## Reduction of 6U/7U with 2 KC₈

In the strict absence of light, benzene (10 mL) was added to a solid mixture of **6U/7U** (0.031 g, 0.015 mmol) and $KC_8$ (0.004 g, 0.03 mmol) at room temperature. There was a gradual precipitation of black insoluble solid. The resultant brown suspension was stirred for two hours, before filtering to yield a very pale-yellow solution. Removal of the volatiles *in vacuo* yielded a pale-yellow oil. NMR spectroscopic analysis of the reaction mixture showed the presence of Tren^TIPS$H_3$[42], as well as unidentifiable resonances. ¹H NMR (C₆D₆, 298 K): 7.52 (s), 7.35 (s), 7.25 (s), 4.05 (s), 3.57 (s), 2.89 (s, 6H, NCH₂CH₂, Tren^TIPSH₃), 2.67 (s), 2.44 (s, 6H, NCH₂CH₂, Tren^TIPSH₃), 2.06 (s), 1.65 (s), 1.36 (s), 1.29 (s), 1.15−0.95 (m, 63H, CH(CH₃)₂ and CH(CH₃)₂, Tren^TIPSH₃), 0.77 (s, 3H, NCH₂CH₂NH, Tren^TIPSH₃) ppm.

## Synthesis and isolation of [U(Tren^TCHS)(μ-SbH)Na(15C5)] (9UNa)

In the strict absence of light, **1Na** (0.073 g, 0.20 mmol) was added in a portion-wise manner to a red stirring slurry of **3U** (0.250 g, 0.20 mmol) in benzene (10 mL) at room temperature. The resultant brown solution was stirred for 16 h, before Celite® was added into the reaction with stirring and then the mixture filtered to yield a dark brown solution. Removal of the volatiles *in vacuo* yielded a brown solid which was dried for two hours. Crystals of **9UNa** suitable for single-crystal X-ray diffraction studies were grown through the slow evaporation of a concentrated Et₂O (10 mL) solution at room temperature; in bulk these crystals appear to be black, but they are actually dark red-black when inspected in detail. The mother liquor was decanted, and the black crystals were washed with Et₂O (2 × 2 mL) and dried *in vacuo* for 30 min to afford **9UNa**. Yield: 0.044 g, 20%. Anal. Calcd for USbNaO₅N₄Si₃C₇₀H₁₃₂: C, 53.32; H, 8.44; N, 3.55%; Found: C, 53.47; H, 8.56; N, 3.18%. ATR-IR v/cm⁻¹: 2912 (m), 2840 (m), 1574 (w), 1442 (m), 1396 (m), 1352 (w), 1290 (w), 1255 (m), 1241 (w), 1189 (m), 1165 (m), 1115 (s), 1103 (m), 1099 (s), 1037 (m), 996 (m), 933 (m), 910 (w), 889 (s), 844 (m), 813 (m), 760 (w), 731 (w), 514 (s), 450 (w), 429 (w). Complex **9UNa** is insoluble in common arene and ethereal solvents after being isolated as a crystalline solid which prohibited the acquisition of solution-state NMR and UV/Vis/NIR spectroscopic data. Sample decomposition precluded the acquisition of Raman spectroscopic data.

## Synthesis and isolation of [U(Tren^TCHS)(μ-SbH)(K(18C6)] (9UK)

In the absence of light, **1K** (0.25 g, 0.50 mmol) was added in a portion-wise manner to a red stirring slurry of **3U** (0.60 g, 0.50 mmol) in

benzene (10 mL) at room temperature. The resultant green/brown solution was stirred for two hours, before Celite® was added into the reaction with stirring and then the mixture filtered to yield a dark brown solution. Removal of the volatiles *in vacuo* yielded a brown solid which was dried for two hours. Crystals of **9UK** suitable for single-crystal X-ray diffraction studies were grown through the slow evaporation of a concentrated $Et_2O$ (10 mL) solution at room temperature; in bulk these crystals appear to be black, but they are actually dark red-black when inspected in detail. The mother liquor was decanted, and the black crystals were washed with $Et_2O$ (2 × 2 mL) and dried *in vacuo* to afford **9UK**. Yield: 0.42 g, 50%. Anal. Calcd for $C_{72}H_{136}KN_4O_6SbSi_3U$: C, 52.83; H, 8.37; N, 3.42%; Found: C, 52.62; H, 8.55; N, 3.38%. ATR-IR ν/cm$^{-1}$: 2909 (s), 2839 (s), 1654 (w), 1444 (m), 1350 (w), 1272 (w), 1248 (w), 1107 (s), 1044 (w), 997 (m), 963 (s), 931 (w), 889 (w), 840 (w), 817 (w), 796 (w), 743 (vs), 743 (s), 678 (w), 552 (m), 521 (m), 452 (w), 437 (w). Raman ν/cm$^{-1}$ (532 nm): 2915 (s), 2833 (s), 2806 (m), 1453 (w), 1427 (w), 1327 (w), 1272 (w), 1178 (vw), 1026 (w), 797 (w), 135 (vs), 50 (s). Complex **9UK** is insoluble in common arene and ethereal solvents after being isolated as a crystalline solid which prohibited the acquisition of solution-state NMR and UV/Vis/NIR spectroscopic data.

### Synthesis and isolation of [K(18C6)(THF)$_2$][U(Tren$^{TCHS}$)(SbH)] (10UK)

A 20 mL glass scintillation vial was charged with a PTFE-coated stirrer bar and **9UK** (0.32 g, 0.20 mmol). At room temperature, THF (10 mL) was added. The resultant dark brown suspension was stirred for 24 h, before Celite® was added into the reaction with stirring and then the mixture filtered to yield a dark brown solution. Removal of the volatiles *in vacuo* yielded a green/brown oily solid which was washed with $Et_2O$ (2 × 5 mL) and dried *in vacuo*, affording **10UK** as green/brown solid which was dried for one hour. Crystals of **10UK** suitable for single-crystal X-ray diffraction studies were grown through the slow evaporation of a concentrated THF (10 mL) solution at room temperature; in bulk these crystals appear to be black, but they are actually dark red-black when inspected in detail. The mother liquor was decanted, and the black crystals were washed with $Et_2O$ (2 × 2 mL) and dried *in vacuo* to afford **10UK**. Yield: 0.21 g, 58%. Anal. Calcd for $C_{80}H_{152}N_4KO_8SbSi_3U$: C, 53.94; H, 8.60; N, 3.15%; Found: C, 53.28; H, 8.79; N, 3.38%. $^1$H NMR (D$_8$-THF, 298 K): δ 3.62 (m, 8H, [C$H_2$C$H_2$]$_2$O), 3.56 (br, 24H, [C$_2H_4$O]$_6$), 2.40 (m, 19H, Cy$H$), 2.19 (s, 6H, NC$H_2$CH$_2$), 1.95 (m, 20H, Cy$H$), 1.79 (m, 8H, [C$H_2$CH$_2$]$_2$O), 1.44 (m, 38H, Cy$H$), 0.75 (m, 22H, Cy$H$), −0.18 (s, 6H, NCH$_2$C$H_2$) ppm. Resonances attributed to the hydrogen atoms on the K-coordinated THF molecules could not be definitively integrated due to overlapping with resonances from the D$_8$-THF solvent needed to acquire the NMR spectrum. $^{29}$Si{$^1$H} NMR (D$_8$-THF, 298 K): δ −73.05 ppm. ATR-IR ν/cm$^{-1}$: 2908 (s), 2839 (s), 1653 (w), 1443 (m), 1350 (w), 1273 (w), 1248 (w), 1109 (vs), 1044 (w), 998 (m), 963 (s), 933 (w), 889 (w), 840 (w), 817 (w), 797 (w), 742 (vs), 678 (w), 552 (m), 521 (m), 454 (w). Raman ν/cm$^{-1}$ (532 nm): 2902 (s), 2835 (s), 2804 (m), 1429 (w), 1265 (vw), 1007 (w), 794 (w), 145 (vs), 77 (s). UV-vis-NIR (THF): λ$_{max}$ (ν/cm$^{-1}$; ε/M$^{-1}$cm$^{-1}$) = 1,219 (8,205; 197), 1,111 (9,003; 293), 964 (10,378; 458), 723 (13,826; 501), 546 (18,295; 802) nm.

### Computational details

Complexes **5U** (the anion component of **5UK**), **6U**, **7U**, **9UNa**, **9UK**, and **10U** (the anion component of **10UK**) were geometry optimised using the atom coordinates from the X-ray crystal structures as the starting points (Supplementary Tables 5–10). Complexes **5U**, **6U**, and **7U** were computed as spin-quintets (four unpaired electrons) and **9UNa**, **9UK**, and **10U** were computed as spin-triplets (two unpaired electrons) consistent with prior studies and also the spin states determined experimentally by magnetometry. No other constraints were imposed on the structures during those geometry optimisations. Single point energy calculations were then performed on the optimised coordinates. The calculations were performed using the Amsterdam Density

Functional (ADF) suite version 2017 with standard convergence criteria[87,88]. The DFT calculations employed Slater type orbital (STO) triple-ζ-plus polarisation all-electron basis sets (from the Dirac and ZORA/TZP database of the ADF suite). Scalar relativistic approaches (spin-orbit neglected) were used within the ZORA Hamiltonian[89–91] for the inclusion of relativistic effects and the local density approximation (LDA) with the correlation potential due to Vosko et al was used in all of the calculations[92]. Generalized gradient approximation (GGA) corrections were performed using the functionals of Becke and Perdew[93,94]. Natural Bond Order (NBO) and Natural Localized Molecular Orbital (NLMO) analyses were carried out with NBO 6.0.19[95]. The Quantum Theory of Atoms in Molecules analysis[53,96,97] was carried out within the ADF program. We quote Nalewajski-Mrozek bond orders since they reproduce expected bond multiplicities reliably in polar heavy atom structures whereas Mayer bond orders for polar bonds often do not conform with chemical intuition[98]. Analytical frequency calculations were carried out within the ADF program. The ADF-GUI (ADFview) was used to prepare the three-dimensional plots of the electron density. We examined the nature of the M = EH (M = U, Th; E = P, As, Sb) using the energy decomposition analysis (EDA) that is incorporated within the ADF code[87,88]. In using this approach, the Bonding Energy ($\Delta E_{bond}$) between two fragments is decomposed into $\Delta E_{bond} = \Delta E_{steric} + \Delta E_{oi}$, where $\Delta E_{steric}$ is the Steric Interaction energy between the two fragments in geometries that are identical to those in the parent molecule and $\Delta E_{oi}$ is the Orbital Interaction in the bonding energy. $\Delta E_{steric}$ comprises the destabilising repulsive interactions between occupied molecular orbitals ($\Delta E_{Pauli}$) and the classical electrostatic interaction ($\Delta E_{elstat}$) between the fragments; $\Delta E_{oi}$ accounts for electron pair bonding, charge transfer, and orbital polarisation. In order to perform the EDA analysis we constructed [M(Tren$^{TIPS}$)]$^+$ and [HSb]$^{2-}$ fragments in identical geometries to those in their respective solid-state structures. The EDA routine requires fragment files from restricted DFT calculations and therefore we used the FRAGOCCUPATIONS keyword to treat the [U(Tren$^{TIPS}$)]$^+$ fragments as if they were unrestricted. It has been recognised that the treatment of fragments in this way is not self-consistent, however it has been shown that this is a fair approximation that can provide significant insight into the chemical bonding between fragments[99].

## Data availability

Crystallographic data for the structures reported in this Article have been deposited at the Cambridge Crystallographic Data Centre, under deposition numbers CCDC 2404124 (**5UNa**), 2404125 (**5UK**), 2404126 (**5UK′**), 2404127 (**6U**), 2404128 (**7U**), 2434801 (**8U**), 2404129 (**9UNa**), 2404130 (**9UK**), and 2404131 (**10UK**). Copies of the data can be obtained free of charge via https://www.ccdc.cam.ac.uk/structures. The spectroscopic, magnetic, and computational data generated in this study are provided in the Supplementary Information (Supplementary Figs. 1–113 and Tables 1–10), and Supplementary Data Files. Source data are provided with this paper.

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

## Acknowledgements

We thank the EPSRC (grants EP/M027015/1, EP/P001386/1, EP/S033181/1, EP/T011289/1, and EP/W029057/1, S.T.L.), ERC (612724, S.T.L.), Deutsche Forschungsgemeinschaft (HA 3466/11-1, C.v.H.), Philipps-Universität Marburg and the University of Manchester (C.v.H. and S.T.L.), including computational resources and associated support services from the Computational Shared Facility. We thank Martin Jennings and Anne Davies at the University of Manchester for carrying out CHN microanalyses. The Alexander von Humboldt Foundation is thanked for a Friedrich Wilhelm Bessel Research Award (S.T.L.).

## Author contributions

R.F.S., J.D., K.D., and N.M. synthesised the complexes and characterised them. J.A.S. recorded and interpreted the magnetic data. A.J.W. collected and refined the crystallographic data. S.T.L. conducted the quantum chemical calculations. J.D., C.v.H., and S.T.L. conceived the research idea, directed the research, analysed and interpreted all the data, and wrote the manuscript with contributions from all the authors.

## Competing interests

The authors declare no competing interests.
