## [Transparent Peer Review file · Nature Communications]

Uranium-stibinidiide, -stibinidene, and -stibido multiple bonds and uranium-nitride formation from multimetallic diuranium-distibene-mediated dinitrogen cleavage

Corresponding Author: Professor Stephen Liddle

Version 0:

Reviewer comments:

Reviewer #1

(Remarks to the Author)

This is a high quality new work from the Du, von Hanisch and Liddle groups utilising the TrenAn framework to expand pnictogen-actinide chemistry. This work on U(IV) follows on from their recent Nature Chem. paper on the thorium analogue with additional magnetic studies and intriguing new dinitrogen activation reactivity observed. The paper is written and data presented to a very high standard and the conclusions are warranted based thereon.

I have only very minor issues and recommend publication once these are addressed.

1. Complex 8U is formed from a mixture of 6U:7U of which the latter is the major component. The authors therefore appear to assume that 7U is the key precursor to 8U but although 6U can be isolated in its pure form by another route I see no evidence that 8U cannot form from this precursor. This would be pertinent to confirm.
2. The mixture 6U:7U was isolated in 26% yield at an 11:89 ratio. Since 6U is highly insoluble but 7U is soluble it is eminently possible that the solution from which the crystals are obtained contains a higher ratio of 7U and that 6U is simply a minor impurity. Alternatively if this is not the case I would be curious if the authors believe (or have evidence that) 6U:7U exists in equilibrium in the reaction medium, which would account for the broad NMR data. The presence of H₂ in the NMR of 6U:7U or exchange with D₂ would confirm this.
3. In the introduction these compounds are described as presenting 'unprecedented U-Sb' bonds. I would argue the groups' recent publication of Th-Sb bonds is precedent and that this wording should be changed.
4. In the paragraph below Figure 4 on page 12 the averages of the bond lengths should be given to only 4 significant figures based on the data for individual bonds.
5. Tables of key IR, UV/vis and Raman bands for the compound set in the SI would be very welcome.

Reviewer #2

(Remarks to the Author)

In the manuscript titled "Uranium-stibinidiide, -stibinidene, and -stibido multiple bonds and uranium-nitride formation from multimetallic diuranium-distibene-mediated dinitrogen cleavage", Sheppard et al. report a series of uranium-antimony bonds; the first of such having been characterized. The synthesis is very interesting and the natural next step from their prior work characterizing similar structures with thorium. The experimental work includes optical and vibrational spectroscopy as well. DFT computations are compared and used to support the central conclusion of the paper, that uranium-antimony bonds are more covalent than their thorium counterparts. This is attributed to both a great spatial overlap between the orbitals and a better energy matching.

Overall, I think the chemistry is very interesting, but I think the theory contribution has several missed opportunities. Given the level of theory and the analysis selected, I think many of conclusions are far too strong. I also think that there needs to be serious effort to include the details from the calculations in the supporting information. When TD-DFT or calculated vibrational spectra are used to assign transitions, one must report in detail what those transitions are. I also think that modeling choices could have been improved to help inform us about the nature of the bonds. I will make suggestions in the

following. Nevertheless, I think the work could be publishable and is certainly of interest to the broader inorganic chemistry community. I would consider supporting publication after major revisions.

I don't think there is much reason to limit yourself to one functional for any DFT study. It is very easy to test functional dependence and support your choice. However, for systems with novel bonds I think it is very helpful to the community to demonstrate this. I also think modern functionals and dispersion-corrected functionals should be tested (more on the latter further on).

The computational details are not sufficient to reproduce the work. I'm most concerned that the TDDFT calculations are reported in the paper but not mentioned in the details. How were these computed? For how many states? To what threshold? What are the values of the calculated transitions? What are the oscillator strengths? SI should also be included with tables showing the details of the transitions (from which orbitals and into which orbitals). The orbital images for the transitions being discussed in the paper should also be included and labeled so a reader can look and see what transitions contribute to each peak.

Was the same functional used in the TDDFT? I would expect one to need a hybrid functional.

From a figure caption and the references, I can see the BP86 functional is used; however, the way this is written in the computational details is unconventional. They mention LDA which could confuse junior researchers reading the paper into thinking LDA calculations were performed when they are not. BP86 is a GGA functional and there is no need to mention LDA in passing.

The authors don't mention if they use a frozen core in the basis set. They also mention they do a single point on the optimized structures but don't say what that is. There is no reason to do a single point if it is at the same level of theory, but no higher level of theory is mentioned.

Which Nalewajski-Mrozek bond order is reported? Three are printed in the output.

What isosurface is used to plot the orbitals?

What was the convergence threshold for these calculations (SCF and geometry)? What is the spin state? Did they only consider high spin? I assume this to be the case, but for systems with multiple bonds, this should be justified. Think about classic metal-oxo bonding. Some cases higher spin is favored leading to an oxygen radical. While the chemistry here is different, the authors may have reasons to suggest only one spin state need be explored, but it should be stated clearly. The authors perform the QTAIM analysis in ADF and report only a table with values but do not mention the position of the critical points. They should include the image with the molecule and the critical points. Then, the critical points in the table should be referred to by their number so one can understand which bond has the specific critical point mentioned. They should also note that these are (3,-1) critical points and include a section in the SI explaining to readers what the metrics in the table mean and how they are interpreting them.

Additional supporting information is also needed for the computed vibrational spectra. For example, on page 10 the authors compare the measured Raman data to computed U=Sb stretching frequencies. This is a metric of bond strength, and it would be helpful to see the values. The calculated frequencies and intensities should be included as SI.

Coordinates should be included as XYZ files in a zipped folder. Taking coordinates from a PDF is frustrating.

Turning to the main paper, the authors should mention the functional when they first discuss DFT. This can be as simple as adding it in parentheses in the text "DFT (BP83) was used..." etc. Similarly, for TDDFT even if the functional is the same. Expert readers will want to know and shouldn't have to flip to the details.

The TDDFT results should be included in Figure 2. If they are used to assign the peaks, we should see how they appear in comparison to experiment.

I find the theory table dense and it is hard to see a trend. I'd encourage the authors to think if there is a plot that would be more informative? What should I get out of the charges presented? I would have been curious to see a comparison (perhaps as a graph) of the important metrics from the U-P, U-As, and U-Sb complexes. In the abstract, the authors state this trend informs them about changes in covalency but I missed this discussion from the theory side.

Likewise, the orbital images could have the atomic percent contributions included in the figures. It would make it easier on the reader to see the percentages near the image.

In Table 1, the U-Sb bond distance is predicted to be too long by DFT. This is likely due to using a functional that doesn't recover dispersion.

For 6U, how was the 3H4 ground state multiplet assigned? Presumably by comparison to the free ion?

In Figure 8, 6U is shown to have sigma bonding and this is discussed in terms of covalency, but earlier in the paper (bottom of page 8 and top of page 9), the authors state that HSb2- is a strong point charge? Wouldn't covalency induce changes in the magnetism that would lead it to deviate from the U(IV) ion electronic structure? How can the authors rationalize this?

Finally, I would ask the authors to address bond strength. The steric bulk on the ligand is essential to make these complexes, but I'm always curious how much that impacts the bond. Would truncating the ligand impact the structure?

Would the bonds get stronger if there was less steric bulk, supporting the author's conclusions about having multiple bonds? Or would the nature of the bond change if the ligands could rotate more strongly?

Is there a reason the authors opted not to perform fragment calculations (EDA-NOCV studies)? I think quantifying the percent electrostatic vs orbitalic across the various complexes and comparing them to the Th systems would be very interesting.

At the top of page 18, the authors use the bond orders to assign bond order. I think this needs to be in combination with a more nuanced discussion of the Kohn Sham orbitals. But I also had a specific question. The authors say that 5U has a bond order of 1.86 and that this implies a triple bond. But in 9UNa and 9UK the bond orders are 1.85 and 1.90 and they assign a double bond. Is there an error here?

The authors could report the delocalization index. This could be used to support their assignment of improved energy degeneracy in U vs Th.

I'm curious if there is an orbital stabilization argument to be made? In this work, the 5f orbitals are involved. Are they more stable in U than in Th? My understanding is that this is a donation from the ligand to empty 5f orbitals. Is it the energy of the 5f orbitals that changes? Is there evidence for this in the calculations? A plot showing the orbital energies with atomic contributions mapped on it could be informative.

I'll add one last thought, and the authors can take it or leave it. I think these are beautiful complexes. Just fantastic. I'm pushing back a bit because I think we can learn a lot from more detailed modeling of these systems. I'd suggest that the theory portion focus on understanding how the bond strengths vary. I'm not sure how much small changes in f-orbital mixing impact bond strength in this case. I'd like to have it quantified so I can learn this. Are the U-Sb bonds meaningfully stronger than a Th-Sb bond? I'd also love to know how these bonds compare to the U-P or U-As systems.

Reviewer #3

(Remarks to the Author)

Liddle and coworkers report in this manuscript the syntheses of uranium-stibido (USbU), -stibinidiide (USb(H)U and USb(H)M, M = Na or K), -distibene (USb₂U), and -stibinidene (USbH) derivatives containing single, double, and pseudo-triple bond interactions. These are the first examples of U-Sb multiple bonding and the reported synthesis provide a very nice comparison complementing the work previously reported by the same authors for Thorium in Nature Chemistry in 2024. Quantum chemical calculations suggest that these uranium-antimony multiple bonds are more covalent than thorium-antimony congeners, due to their superior spatial and energy matching of uranium and antimony frontier orbitals. This is a step forward into understanding bonding in actinide elements.

The work is carried out at the usual high standards of the Liddle group and makes a nice contribution well suited for Nature Communications.

I recommend publication after the following points have been taken into consideration.

i) The authors should clearly state in the text that ¹H and ²⁹Si{¹H} Nuclear Magnetic Resonance (NMR) spectra of 5UNa, 5UK, and 5UK' show the same chemical shifts and are exactly the same (I think this is the case but the spectra are not overlapped so it is difficult to be certain) which is a proof that cations are not bound. (The authors mention that for UV data: suggesting the presence of fully separated ion pairs in solution and hence a common [U(TrenTIPS)]₂(μ-Sb)]⁻ anion for all three complexes) but I think ¹H NMR is a stronger tool to define this.

ii) The observed difference in magnetic data (χ vs T) for 5UK' compared to 5UNa, 5UK is quite puzzling considering the only difference is the counterion (unless in 5UNa, 5UK the counterion is innersphere). How can this be rationalized? A comment should be provided.

iii) I am not convinced measuring magnetic data of a mixture of compounds can really bring much information (besides suggesting the presence of U(IV) in both complexes) and one should be careful in the presentation. In my opinion talking about "magnetic moment" of a mixture of two quite different species 6U and 7U has no meaning and should be avoided. Can the authors be really sure that the composition of the bulk compound is the same as in the crystal without PXRD data? I guess one could correct the data of 7U by subtracting the component due to the 11% 6U but one has to be sure of the real composition. The 11:89 ratio in 6U/7U (prepared from 1K) is solely XRD-derived. Stoichiometry of this reaction should be further clarified by supplying readers with ¹H NMR of the crude reaction mixture.

iv) More importantly the following point should be addressed:

The reduction of 6U/7U with KC8, under N₂ affording diuranium dinitride complex [U(TrenTIPS)(μ-NK)]₂ (8U) in 19% yield, is very interesting but perhaps too preliminary to be presented in the context of this paper without further studies. The role of the bound Sb₂ fragment should be proven or at the very least the possibility of reduction being the result of a transient U(II) generated by reduction of the precursor 4U should not be ruled out, but proposed as an alternative mechanism. Have the authors performed the reduction of the precursor 4U in exactly the same conditions that were used for the isolation of 8U? This should be done even for presenting the result as preliminary. Comparison of reaction mixtures should allow to see if 8U can be formed in the absence of the Sb₂ group. The authors mention reduction of [U(TrenTIPS)] as failing to give a nitride or reaction with N₂ but there is no experimental section or literature reference supporting this statement. More details about the reduction trials with excess KC8 should be given.

What happens if the reduction is performed with lower amounts of KC8 (1 or 2 equivalents).

The presence of a transient U(II) species that could be active in N₂ reduction is definitely possible, but I don't think the authors have enough evidence that rules out the formation of this species in the absence of Sb₂.

Moreover, a completely different pathway could also be at play: indeed the Sb₂ unit could act as an electron reservoir for the

cooperative N₂ reduction without the need of invoking a U(II) intermediate.

Comparison of characterization data for 8U and previously reported nitride should be presented in SI (what data were used ? X-ray, NMR?). In particular in the absence of labeling experiments the formation of a nitride should be evidenced by NMR comparisons etc.

I think the synthesis of the first complexes containing U-Sb multiple bond is already ground for publication in Nature Communications, but if the authors want to include this unsuspected and the very unusual reactivity as a result of Sb presence they should dig a little more into it or leave this for a future publication.

v) Authors state numerous times that 6U is insoluble in common arene and ethereal solvents, however they make no such remark about 7U. That should facilitate 6U/7U separation without the need for fractional co-crystallisation and, in turn, would give access to pure 7U to allow to open up more of its reduction chemistry. Please comment on this.

VI) It is reported that reduction of 6U:7U with KC8/cryptand gives 5UK'. Based on the stoichiometric considerations, is more likely that it is only 6U component that is responsible for this reactivity mode and gives 5UK' through the removal of 1/2 H₂. That could be probed by independently reducing 6U (with that authors are able to isolate in the clean form) with KC8/cryptand.

VII) The stark difference in 1Na/1K reactivity with 4U is quite remarkable. Have the author an explanation for this divergent reactivity?

In conclusion I think this work is perfectly suited for publication in Nature Communications and I recommend publication. Most of the above comments do not require significant experimental effort but just some polishing of the presentation.

Version 1:

Reviewer comments:

Reviewer #1

(Remarks to the Author)

I am happy that the authors have fully addressed my comments.

Reviewer #2

(Remarks to the Author)

I thank the reviewers for their thorough response to my questions. I think it is clear from my initial reading that I got quite confused about what DFT work had been performed, in particular with respect to TD-DFT or the lack thereof. I think the manuscript is more clear now. The addition of the EDA results was great, and I helped me understand the chemistry being described more deeply. I hope my comments have been helpful to the reviewers as they were offered with that in mind. I recommend accepting the paper in its current form.

Reviewer #3

(Remarks to the Author)

The authors have replied carefully to most of the points raised by the referees in the previous round of reviews. I have only one point that still need addressing.

I agree that the additional experiments corroborate the hypothesis that the presence of bound Sb₂ in 7-U is important to the isolation of the nitride.

However, I find that the discussion advocating a transient U(II) (there are no precedents of N₂ activation by U(II)) is not supported by spectroscopic or computational evidence and should be kept to a minimum and associated to another hypothesis.

Considering the paragraph:

“We surmise that a U(II) oxidation state, at a minimum, would be required, since trivalent [U(TrenTIPS)] is stable under N₂ for weeks. We note that [U(TrenTIPS)] does not seem to be reduced when placed in the presence of excess K, indeed it is prepared in the presence of excess Kmirror or KC₈,⁴² and so perhaps the Sb₂-unit provides an alternative, energetically more accessible indirect route to reduction of U that is not otherwise normally directly feasible.”

The reported experiments cannot either prove the presence of a transient U(II) or rule out a highly reactive U(III) intermediate resulting from the interaction of USb₂-U with N₂ and these hypotheses should in my opinion both be presented as possible alternatives.

Once this point has been dealt with the paper is ready for publication.

REVIEWER COMMENTS – Round 1

Reviewer #1 (Remarks to the Author):

This is a high quality new work from the Du, von Hanisch and Liddle groups utilising the TrenAn framework to expand pnictogen-actinide chemistry. This work on U(IV) follows on from their recent Nature Chem. paper on the thorium analogue with additional magnetic studies and intriguing new dinitrogen activation reactivity observed. The paper is written and data presented to a very high standard and the conclusions are warranted based thereon. I have only very minor issues and recommend publication once these are addressed.

RESPONSE: We thank the reviewer for their support and thoughts and address their queries below.

1. Complex 8U is formed from a mixture of 6U:7U of which the latter is the major component. The authors therefore appear to assume that 7U is the key precursor to 8U but although 6U can be isolated in its pure form by another route I see no evidence that 8U cannot form from this precursor. This would be pertinent to confirm.

RESPONSE: The reviewer raises a good point. We have therefore tested the reduction of pure **6U** with KC_8 , with and without 2.2.2-cryptand, and do not isolate any compounds other than $[U(Tren^{TIPS})]$ and $Tren^{TIPS}H_3$. This demonstrates that it is **7U** that is the vehicle for N_2 reduction. We have added the additional experiments to the experimental section and short discussions to the results and discussion sections.

2. The mixture 6U:7U was isolated in 26% yield at an 11:89 ratio. Since 6U is highly insoluble but 7U is soluble it is eminently possible that the solution from which the crystals are obtained contains a higher ratio of 7U and that 6U is simply a minor impurity. Alternatively if this is not the case I would be curious if the authors believe (or have evidence that) 6U:7U exists in equilibrium in the reaction medium, which would account for the broad NMR data. The presence of H_2 in the NMR of 6U:7U or exchange with D_2 would confirm this.

RESPONSE: We have focussed on what crystallises because the NMR of **6U:7U** is broad so examining the mother liquor is not informative and so does not in itself enable us to determine the ratio. We would say, however, that the same situation occurs with the previously published Th analogue where we found excellent agreement between the NMR and XRD occupancy data. We do not believe that it is plausible for there to be an equilibrium between **6U** and **7U** because the U:Sb ratios in **6U** and **7U** are different and we do not find any evidence for H_2 formation when crystalline **6U:7U** is dissolved in a sealed NMR tube – there would have to be an astonishing sequence of bond cleavage and formation events for there to be a persistent equilibrium between **6U** and **7U**. That said it is entirely possible that *in situ* formed “[$(Tren^{TIPS})U(SbH_2)$]” reacts with H_2Sb^{1-} to give SbH_3 and “[$(Tren^{TIPS})U(SbH_2)$] $^{1-}$ ” which if deprotonated again could combine with the U part of **4U** (nicely eliminating $MBPh_4$) to give **6U** or through further deprotonation/dehydrocoupling produce **7U**. We note that the prevalence of **6U** for $M = Na$ vs **7U** with $M = K$ would be entirely consistent with the Sb-M linkage being more polar and hence reactive (at Sb) for K compared to Na. The reviewer has touched on an important fundamental issue here so we have modified Figure 1 (reaction scheme) to reflect the likely extrusion of SbH_3 .

3. In the introduction these compounds are described as presenting 'unprecedented U-Sb' bonds. I would argue the groups' recent publication of Th-Sb bonds is precedent and that this wording should be changed.

RESPONSE: Our phrase conveyed that the linkages were unprecedented for U, which is correct, but we take the point and have no issue with removing the phrase and have done so.

4. In the paragraph below Figure 4 on page 12 the averages of the bond lengths should be given to only 4 significant figures based on the data for individual bonds.

RESPONSE: Good spot, corrected to match all other examples.

5. Tables of key IR, Uv/vis and Raman bands for the compound set in the SI would be very welcome.

RESPONSE: We had already listed data in numerical form in the experimental section and have now added tabulated data in the SI.

Reviewer #2 (Remarks to the Author):

In the manuscript titled “Uranium-stibinidiide, -stibinidene, and -stibido multiple bonds and uranium-nitride formation from multimetallic diuranium-distibene-mediated dinitrogen cleavage”, Sheppard et al. report a series of uranium-antimony bonds; the first of such having been characterized. The synthesis is very interesting and the natural next step from their prior work characterizing similar structures with thorium. The experimental work includes optical and vibrational spectroscopy as well. DFT computations are compared and used to support the central conclusion of the paper, that uranium-antimony bonds are more covalent than their thorium counterparts. This is attributed to both a great spatial overlap between the orbitals and a better energy matching.

Overall, I think the chemistry is very interesting, but I think the theory contribution has several missed opportunities. Given the level of theory and the analysis selected, I think many of conclusions are far too strong. I also think that there needs to be serious effort to include the details from the calculations in the supporting information. When TD-DFT or calculated vibrational spectra are used to assign transitions, one must report in detail what those transitions are. I also think that modeling choices could have been improved to help inform us about the nature of the bonds. I will make suggestions in the following. Nevertheless, I think the work could be publishable and is certainly of interest to the broader inorganic chemistry community. I would consider supporting publication after major revisions.

RESPONSE: We thank the reviewer for their support and thoughts and address their queries below.

I don't think there is much reason to limit yourself to one functional for any DFT study. It is very easy to test functional dependence and support your choice. However, for systems with novel bonds I think it is very helpful to the community to demonstrate this. I also think modern functionals and dispersion-corrected functionals should be tested (more on the latter further on).

RESPONSE: The reviewer raises an interesting point. We addressed these points at the beginning of the computational section but are happy to discuss them here again. There is of course a time-honoured tradition of testing various functionals when using DFT to study molecules. We have no issue with that, but one thing we think it is important to do between papers is to maintain a consistent way of doing our DFT calculations because over time a great benefit accrues from having all papers being internally consistent. We have consistently used BP86 and whilst it isn't a hybrid functional it does a good job of describing relatively coarse metrics such as bond orders, and bond breakdowns. We can say that based on experimentally validated

calculations, which is when we have computed NMR studies with the hybrid functional B3LYP the NBO/NLMO outputs on those NMR validated calculations are in good agreement with NBO/NLMO outputs from BP86 calculations. We would certainly say that for spectroscopic properties that are very sensitive to spin orbit coupling then hybrids are necessary, but for bond orders and the like it isn't mandatory. The dispersion comment is also interesting. Generally speaking, when using ADF we have over the course of computing many U and Th molecules often found that we do not need to explicitly include dispersion in the input file to obtain equilibrium structures that are in good agreement with solid state structures. However, we certainly acknowledge that sometimes dispersion corrections are necessary, see later.

The computational details are not sufficient to reproduce the work. I'm most concerned that the TDDFT calculations are reported in the paper but not mentioned in the details. How were these computed? For how many states? To what threshold? What are the values of the calculated transitions? What are the oscillator strengths? SI should also be included with tables showing the details of the transitions (from which orbitals and into which orbitals). The orbital images for the transitions being discussed in the paper should also be included and labeled so a reader can look and see what transitions contribute to each peak.

RESPONSE: We think there must be a misunderstanding here because we did not do any TD-DFT calculations on this occasion. That was due to the f^n natures of these complexes. What we did was observe that, putting f-f transitions in the NIR region to one side, experimental spectra of these U complexes are very similar to those of the Th analogues but red-shifted. Since we had done the TD-DFT on the Th complexes we handrailed from the Th to U. So that is why there are no TD-DFT details.

Was the same functional used in the TDDFT? I would expect one to need a hybrid functional. From a figure caption and the references, I can see the BP86 functional is used; however, the way this is written in the computational details is unconventional. They mention LDA which could confuse junior researchers reading the paper into thinking LDA calculations were performed when they are not. BP86 is a GGA functional and there is no need to mention LDA in passing.

RESPONSE: As explained above there aren't any TD-DFT calculations in this submission. Regarding the other points, the computational details are written out the way they are due to the way commands are constructed in the ADF run.in file, which in the functional section is written as:

```
XC
LDA VWN
GGA Becke Perdew
END
```

That is why we write it out as LDA with VWN correlation potential and then GGA using BP86 functional.

The authors don't mention if they use a frozen core in the basis set. They also mention they do a single point on the optimized structures but don't say what that is. There is no reason to do a single point if it is at the same level of theory, but no higher level of theory is mentioned.

RESPONSE: We didn't use the phrase frozen core but in the computational details section we stated "*The DFT calculations employed Slater type orbital (STO) triple- ζ -plus polarisation all-electron basis sets (from the Dirac and ZORA/TZP database of the ADF suite)*" where the phrase 'all-electron' means there is not a frozen core. Regarding the single point energy comment, we were taught to geometry optimise a structure and then run a separate single point energy

calculation. This probably originates from a time when geometry and single point calculations used slightly different convergence criteria but we keep the habit because it helps us to organise our work.

Which Nalewajski-Mrozek bond order is reported? Three are printed in the output.

RESPONSE: We always use the values in the third column, the 'N-M (3)' which if we didn't tell the program to 'print-all' would be the default N-M bond order that would be printed in the output.

What isosurface is used to plot the orbitals?

RESPONSE: As part of assessing dispersion effects (see below) we have replotted all the DFT figures and for MOs, NBOs, and NLMOs have consistently plotted them at the 0.04 a.u. isosurface level which is now stated in figures.

What was the convergence threshold for these calculations (SCF and geometry)?

RESPONSE: As we stated in the Computational Details section they were the standard for ADF, which is 1e-6.

What is the spin state? Did they only consider high spin? I assume this to be the case, but for systems with multiple bonds, this should be justified. Think about classic metal-oxo bonding. Some cases higher spin is favored leading to an oxygen radical. While the chemistry here is different, the authors may have reasons to suggest only one spin state need be explored, but it should be stated clearly.

RESPONSE: We ran triplet and quintet spin states for f^2 and $2 \times f^2$ complexes, respectively. This is because we've never found a system that didn't have singlet states lying many kcal/mol above the triplet states (per ion), but also because those spin states are consistent with the spin states of the molecules as determined experimentally by SQUID magnetometry. The reviewer is right to highlight this matter so we have addressed it in the Computational Details section.

The authors perform the QTAIM analysis in ADF and report only a table with values but do not mention the position of the critical points. They should include the image with the molecule and the critical points. Then, the critical points in the table should be referred to by their number so one can understand which bond has the specific critical point mentioned. They should also note that these are (3,-1) critical points and include a section in the SI explaining to readers what the metrics in the table mean and how they are interpreting them.

RESPONSE: It was stated in the manuscript that we were referring to the U-Sb (3,-1) critical points but the reviewer is correct that this was missing from the header of the relevant Supplementary Table 10 so this has been corrected. Given we are only concerned with the U-Sb bond critical points it would be excessive to produce figures with cross-referenced BCPs that the reader has to dig through when we are only concerned with one (3,-1) BCP per molecule so with the matter already defined in the manuscript and now corrected in the SI that is sufficient. In terms of explaining QTAIM we have cited relevant Bader references to direct the interested reader, which is normal practice. Otherwise, every paper in existence would have to explain how every technique and method works every time, which would be excessive.

Additional supporting information is also needed for the computed vibrational spectra. For example, on page 10 the authors compare the measured Raman data to computed U=Sb

stretching frequencies. This is a metric of bond strength, and it would be helpful to see the values. The calculated frequencies and intensities should be included as SI.

RESPONSE: Since the experimental data are already listed in the Experimental Section with plots in the SI we have added plots of the computed IR spectra into the SI for direct comparison. Key metrics were already called-out in the manuscript and are now tabulated in additional tables in the SI.

Coordinates should be included as XYZ files in a zipped folder. Taking coordinates from a PDF is frustrating.

RESPONSE: We have moved the coordinated and energies to separate xyz files.

Turning to the main paper, the authors should mention the functional when they first discuss DFT. This can be as simple as adding it in parentheses in the text “DFT (BP83) was used...” etc. Similarly, for TDDFT even if the functional is the same. Expert readers will want to know and shouldn't have to flip to the details.

RESPONSE: This has been done.

The TDDFT results should be included in Figure 2. If they are used to assign the peaks, we should see how they appear in comparison to experiment.

RESPONSE: As stated above we didn't do any TD-DFT in this study.

I find the theory table dense and it is hard to see a trend. I'd encourage the authors to think if there is a plot that would be more informative? What should I get out of the charges presented? I would have been curious to see a comparison (perhaps as a graph) of the important metrics from the U-P, U-As, and U-Sb complexes. In the abstract, the authors state this trend informs them about changes in covalency but I missed this discussion from the theory side.

RESPONSE: We considered listing all U then all Th but that isn't as effective, and Table 1 is only summarising salient data with the information split up in different tables in the SI. The relevant trends that the reviewer refers to were and remain in the discussion section, paragraph 6.

Likewise, the orbital images could have the atomic percent contributions included in the figures. It would make it easier on the reader to see the percentages near the image.

RESPONSE: This has been done.

In Table 1, the U-Sb bond distance is predicted to be too long by DFT. This is likely due to using a functional that doesn't recover dispersion.

RESPONSE: We thank the reviewer for the challenge; it's a fair point. We have looked at dispersion corrected geometry optimisations and obtain a mixed set of outcomes. For **5U** with or without dispersion we end up with U-Sb distances of -0.04 \AA and $+0.04/+0.05 \text{ \AA}$, respectively, compared to the SC-XRD structures. We prefer to use slightly longer U-Sb distances since too short could end up disproportionately enhancing the bonding and we wouldn't want to overclaim so we keep **5U** the same. For **6U** with dispersion we obtain U-Sb distances that are $+0.05/+0.01 \text{ \AA}$ which is better than before so we have now used dispersion corrected data for **6U**. For **7U**, with dispersion U-Sb distances are $-0.07/-0.10 \text{ \AA}$ to experiment and without they are $+0.02/+0.07 \text{ \AA}$ so without dispersion is closer overall so we keep the same. For **9UNa**, we obtain a dispersion

corrected U-Sb that is +0.07 Å which is closer than without dispersion so now we use the dispersion data. Likewise, for **9UK** we have +0.08 Å with dispersion which is improved on without dispersion so we use dispersion corrected now. Lastly, for **10U** with dispersion we obtain a U-Sb distance that is +0.01 Å compared to +0.02 Å without dispersion, so we use the marginally improved dispersion corrected now. We have consequently updated all the data listings and figures and discussion with only minor changes to quoted values.

For 6U, how was the 3H4 ground state multiplet assigned? Presumably by comparison to the free ion?

RESPONSE: The U(IV) ion will be 3H_4 from Russell Saunders coupling scheme, and then the point we were making is that strongly donating 'axial' ligands change the ground state, due to over-rigid symmetry effects, from singlet to triplet which can be evidenced experimentally by analysis of the low temperature magnetic data.

In Figure 8, 6U is shown to have sigma bonding and this is discussed in terms of covalency, but earlier in the paper (bottom of page 8 and top of page 9), the authors state that HSb²⁻ is a strong point charge? Wouldn't covalency induce changes in the magnetism that would lead it to deviate from the U(IV) ion electronic structure? How can the authors rationalize this?

RESPONSE: The language used was correct in each individual context but we agree it could be confusing so this comes down to a choice of language, so we have changed instances of 'strong point charge' to 'strong donor'.

Finally, I would ask the authors to address bond strength. The steric bulk on the ligand is essential to make these complexes, but I'm always curious how much that impacts the bond. Would truncating the ligand impact the structure? Would the bonds get stronger if there was less steric bulk, supporting the author's conclusions about having multiple bonds? Or would the nature of the bond change if the ligands could rotate more strongly?

RESPONSE: Our prior experience is that when ligand sterics are truncated in computational models then as expected M-L distances get shorter and yes in other studies multiple bonds became more fully developed; there can be other knock-on issues as well since truncation normally impacts the electronic as well as steric nature of the donor atoms in ancillary ligands, e.g. if we turn a SiPr₃ into SiH₃, or even SiMe₃, then the silyl-group interaction with the Tren amides changes significantly, the amide is now of a different donor strength, and then the metal is directly impacted which impacts the linkage of interest from an electronic perspective as well as any sterics. It is for that reason that we normally probe full molecules without truncation as a default.

Is there a reason the authors opted not to perform fragment calculations (EDA-NOCV studies)? I think quantifying the percent electrostatic vs orbitalic across the various complexes and comparing them to the Th systems would be very interesting.

RESPONSE: We were originally mindful of paper length but are pleased to now include EDA analysis of U=PH, U=AsH, U=SbH, Th=PH, Th=AsH, and Th=SbH linkages since we have all of those structures made and characterised. They are also good to study through the fragment approach since splitting MSbM linkages is problematic. The EDA routine requires fragment files from restricted DFT calculations. This is obviously not an issue for Th, but for U it requires use of the FRAGOCCUPATIONS keyword to treat the 5f² [U(Tren^{TIPS})]⁺ fragments as if they were unrestricted. It has been recognised that the treatment of fragments in this way is not self-consistent, however it has also been shown that this is a fair approximation that can provide

significant insight into the chemical bonding between fragments. We are delighted to say that the EDA analysis is consistent with everything stated before and indeed reinforces our statements as well as demonstrating several interesting trends so thank the reviewer for the prompt. The new EDA results are now included in the manuscript and SI.

At the top of page 18, the authors use the bond orders to assign bond order. I think this needs to be in combination with a more nuanced discussion of the Kohn Sham orbitals. But I also had a specific question. The authors say that 5U has a bond order of 1.86 and that this implies a triple bond. But in 9UNa and 9UK the bond orders are 1.85 and 1.90 and they assign a double bond. Is there an error here?

RESPONSE: As we wrote in the full text, our statements pertained to a pseudo molecular orbital triple bond that in a Lewis description would be a double bond so there is not any contradiction there.

The authors could report the delocalization index. This could be used to support their assignment of improved energy degeneracy in U vs Th.

RESPONSE: As we mentioned earlier, we tend to use the 'N-M (3)' BI to be consistent with other studies and also as stated in the Computational Details section due to the reasons Autschbach gives in the reference we quoted.

I'm curious if there is an orbital stabilization argument to be made? In this work, the 5f orbitals are involved. Are they more stable in U than in Th? My understanding is that this is a donation from the ligand to empty 5f orbitals. Is it the energy of the 5f orbitals that changes? Is there evidence for this in the calculations? A plot showing the orbital energies with atomic contributions mapped on it could be informative.

RESPONSE: The reviewer touches on a very interesting and also complex, inter-related issue that is not readily disentangled. Complicating matters from the off is that some of the complexes are anions and others are neutral. Then it is clear that there is a balance of not two but likely several U valence orbitals (5f, 6p, 6d, 7s, 7p). There are too many variables. However, with the EDA section now added we feel we have addressed the spirit of this query.

I'll add one last thought, and the authors can take it or leave it. I think these are beautiful complexes. Just fantastic. I'm pushing back a bit because I think we can learn a lot from more detailed modeling of these systems. I'd suggest that the theory portion focus on understanding how the bond strengths vary. I'm not sure how much small changes in f-orbital mixing impact bond strength in this case. I'd like to have it quantified so I can learn this. Are the U-Sb bonds meaningfully stronger than a Th-Sb bond? I'd also love to know how these bonds compare to the U-P or U-As systems.

RESPONSE: We had already discussed the U=PH, U=AsH, and U=SbH series in the discussion section but now that is pleasingly strengthened, and confirmed, by the EDA calculations. The answer is clear, yes the U-Sb bonds are stronger than the Th-Sb ones, just, and the U vs Th stability gap widens as you go up Group 15.

Reviewer #3 (Remarks to the Author):

Liddle and coworkers report in this manuscript the syntheses of uranium-stibido (USbU), -stibinide (USb(H)U and USb(H)M, M = Na or K), -distibene (USb₂U), and -stibinidene (USbH) derivatives containing single, double, and pseudo-triple bond interactions. These are the first

examples of U-Sb multiple bonding and the reported synthesis provide a very nice comparison complementing the work previously reported by the same authors for Thorium in Nature Chemistry in 2024. Quantum chemical calculations suggest that these uranium-antimony multiple bonds are more covalent than thorium-antimony congeners, due to their superior spatial and energy matching of uranium and antimony frontier orbitals. This is a step forward into understanding bonding in actinide elements. The work is carried out at the usual high standards of the Liddle group and makes a nice contribution well suited for Nature Communications. I recommend publication after the following points have been taken into consideration.

RESPONSE: We thank the reviewer for their support and thoughts and address their queries below.

i) The authors should clearly state in the text that ^1H and $^{29}\text{Si}\{^1\text{H}\}$ Nuclear Magnetic Resonance (NMR) spectra of 5UNa, 5UK, and 5UK' show the same chemical shifts and are exactly the same (I think this is the case but the spectra are not overlapped so it is difficult to be certain) which is a proof that cations are not bound. (The authors mention that for UV data: suggesting the presence of fully separated ion pairs in solution and hence a common $[\{\text{U}(\text{TrenTIPS})\}_2(\mu\text{-Sb})]$ -anion for all three complexes) but I think ^1H NMR is a stronger tool to define this.

RESPONSE: The reviewer makes a good point. Considering small variances will occur with paramagnetic samples, the very similar ^1H and ^{29}Si chemical shifts of the TrenU components in solution for all three compounds do indeed suggest a common species as also suggested by the UV/Vis/NIR data. We have therefore revised the text to clearly point that out.

ii) The observed difference in magnetic data (χ vs T) for 5UK' compared to 5UNa, 5UK is quite puzzling considering the only difference is the counterion (unless in 5UNa, 5UK the counterion is inner sphere). How can this be rationalized? A comment should be provided.

RESPONSE: The reviewer raises a good point. Each complex has a different cation component, and thus the crystal packing will vary, which in turn will influence the equilibrium intermolecular interactions, bond lengths and angles, and crystal field; it is well known that small changes in those properties can result in large changes to observed magnetic properties. As an example, even solvate changes can significantly affect observed magnetism; we have added a point addressing this matter along with a supporting reference.

iii) I am not convinced measuring magnetic data of a mixture of compounds can really bring much information (besides suggesting the presence of U(IV) in both complexes) and one should be careful in the presentation. In my opinion talking about "magnetic moment " of a mixture of two quite different species 6U and 7U has no meaning and should be avoided. Can the authors be really sure that the composition of the bulk compound is the same as in the crystal without PXRD data? I guess one could correct the data of 7U by subtracting the component due to the 11% 6U but one has to be sure of the real composition. The 11:89 ratio in 6U/7U (prepared from 1K) is solely XRD-derived. Stoichiometry of this reaction should be further clarified by supplying readers with ^1H NMR of the crude reaction mixture.

RESPONSE: We agree with the reviewer that the data for the mixture of **6U:7U** serve the purpose of demonstrating the presence of U(IV) ions, but are of limited further use, and as such they are presented on a they-are-what-they-are basis only. We agree that use of the phrase 'magnetic moment' isn't optimal so have replaced that phrase with 'magnetic response'. PXRD is quite problematic given the air sensitivity of the samples, also the NMR spectrum of clean material is broad so the reaction mother liquor is not any more informative. However, the result is analogous

to our prior work with Th where the XRD/NMR ratios were consistent with each other so whilst we cannot be 100% certain of the 11:89 ratio it isn't out of place and given we're only using the magnetism to confirm U(IV) it isn't essential to verify any further.

iv) More importantly the following point should be addressed:

The reduction of 6U/7U with KC8, under N2 affording diuranium dinitride complex $[\{U(\text{TrenTIPS})(\mu\text{-NK})\}_2]$ (8U) in 19% yield, is very interesting but perhaps too preliminary to be presented in the context of this paper without further studies. The role of the bound Sb2 fragment should be proven or at the very least the possibility of reduction being the result of a transient U(II) generated by reduction of the precursor 4U should not be ruled out, but proposed as an alternative mechanism. Have the authors performed the reduction of the precursor 4U in exactly the same conditions that were used for the isolation of 8U ? This should be done even for presenting the result as preliminary. Comparison of reaction mixtures should allow to see if 8U can be formed in the absence of the Sb2 group. The authors mention reduction of $[\{U(\text{TrenTIPS})\}]$ as failing to give a nitride or reaction with N2 but there is no experimental section or literature reference supporting this statement. More details about the reduction trials with excess KC8 should be given.

RESPONSE: The reviewer raises a fair point. We have therefore tested the reduction of pure **6U** with KC₈, with and without 2.2.2-cryptand, and do not isolate any compounds other than $[U(\text{Tren}^{\text{TIPS}})]$ and $\text{Tren}^{\text{TIPS}}\text{H}_3$. This demonstrates that it is **7U** that is the vehicle for N₂ reduction. We have also examined further reduction reactions of $[U(\text{Tren}^{\text{TIPS}})]$ and **4U** and can only evidence formation of $[U(\text{Tren}^{\text{TIPS}})]$ or $\text{Tren}^{\text{TIPS}}\text{H}_3$ proligand. We have also tested reduction of **6U/7U** with half the amount of KC₈ and find only $\text{Tren}^{\text{TIPS}}\text{H}_3$. Those results demonstrate: (i) that Sb_2^{2-} rather than HSb^{2-} is the vehicle for N₂ reduction; (ii) that the combination of U and Sb is important to effect N₂ reduction at all; (iii) the reductant (K) is also clearly playing a vital role. We have added the additional experiments to the experimental section and short discussions to the results and discussion sections.

What happens if the reduction is performed with lower amounts of KC8 (1 or 2 equivalents). The presence of a transient U(II) species that could be active in N2 reduction is definitely possible, but I don't think the authors have enough evidence that rules out the formation of this species in the absence of Sb2.

RESPONSE: we addressed this point in our response above but would also say that observation (iii) is consistent with the proposed importance of U(II) to N₂ reduction and have also made that point in the discussion now.

Moreover, a completely different pathway could also be at play: indeed the Sb2 unit could act as an electron reservoir for the cooperative N2 reduction without the need of invoking a U(II) intermediate.

Comparison of characterization data for 8U and previously reported nitride should be presented in SI (what data were used ? X-ray, NMR?). In particular in the absence of labeling experiments the formation of a nitride should be evidenced by NMR comparisons etc.

RESPONSE: The reviewer makes a fair point which we now acknowledge in the discussion. Noting that **8U** is a poorly soluble dimer that precipitates from arene and alkane solvents (and that decomposes in ethers) we have now added a new crystal structure of **8U** and provided new ATR-IR data with a comparison to prior Nujol IR data.

I think the synthesis of the first complexes containing U-Sb multiple bond is already ground for

publication in Nature Communications, but if the authors want to include this unexpected and the very unusual reactivity as a result of Sb presence they should dig a little more into it or leave this for a future publication.

RESPONSE: Given the extra results that are now included on this matter we prefer to keep the result since irrespective of any details the overall result is we think notable enough to be published without delay.

v) Authors state numerous times that 6U is insoluble in common arene and ethereal solvents, however they make no such remark about 7U. That should facilitate 6U/7U separation without the need for fractional co-crystallisation and, in turn, would give access to pure 7U to allow to open up more of its reduction chemistry. Please comment on this.

RESPONSE: What we observe is that species are in solution, then they are crystallised, then they don't redissolve and since 6U is effectively bound up in a matrix of 7U whilst we agree with the reviewer's point it's not feasible in practice.

VI) It is reported that reduction of 6U:7U with KC8/cryptand gives 5UK'. Based on the stoichiometric considerations, is more likely that it is only 6U component that is responsible for this reactivity mode and gives 5UK' through the removal of 1/2 H₂. That could be probed by independently reducing 6U (with that authors are able to isolate in the clean form) with KC8/cryptand.

RESPONSE: The reviewer raises a good point worth checking. As we described above to address the N₂ activation, reduction of pure 6U gives Tren^{TIP}S₃ proligand, emphasising that the chemistry is 7U-centred.

VII) The stark difference in 1Na/1K reactivity with 4U is quite remarkable. Have the author an explanation for this divergent reactivity?

RESPONSE: The reviewer makes a good point. As we mentioned in response to a query from R1 we note that the prevalence of 6U for M = Na vs 7U with M = K would be entirely consistent with the Sb-M linkage being more polar and hence basic (at Sb) for K compared to Na. A comment on this matter has now been added to the discussion section.

In conclusion I think this work is perfectly suited for publication in Nature Communications and I recommend publication. Most of the above comments do not require significant experimental effort but just some polishing of the presentation.

RESPONSE: We thank the reviewer for their support and thoughts.

REVIEWER COMMENTS – Round 2

Reviewer #1 (Remarks to the Author):

I am happy that the authors have fully addressed my comments.

RESPONSE: We thank the reviewer for their support, thoughts, and time – the result is an improved manuscript.

Reviewer #2 (Remarks to the Author):

I thank the reviewers for their thorough response to my questions. I think it is clear from my initial reading that I got quite confused about what DFT work had been performed, in particular with respect to TD-DFT or the lack thereof. I think the manuscript is more clear now. The addition of the EDA results was great, and helped me understand the chemistry being described more deeply. I hope my comments have been helpful to the reviewers as they were offered with that in mind. I recommend accepting the paper in its current form.

RESPONSE: We thank the reviewer for their support, thoughts and time – the result is an improved manuscript.

Reviewer #3 (Remarks to the Author):

The authors have replied carefully to most of the points raised by the referees in the previous round of reviews. I have only one point that still need addressing.

I agree that the additional experiments corroborate the hypothesis that the presence of bound Sb_2 in 7-U is important to the isolation of the nitride.

However, I find that the discussion advocating a transient U(II) (there are no precedents of N_2 activation by U(II)) is not supported by spectroscopic or computational evidence and should be kept to a minimum and associated to another hypothesis.

Considering the paragraph:

“We surmise that a U(II) oxidation state, at a minimum, would be required, since trivalent [U(TrenTIPS)] is stable under N_2 for weeks. We note that [U(TrenTIPS)] does not seem to be reduced when placed in the presence of excess K, indeed it is prepared in the presence of excess Kmirror or KC_8 ,⁴² and so perhaps the Sb_2 -unit provides an alternative, energetically more accessible indirect route to reduction of U that is not otherwise normally directly feasible.”

The reported experiments cannot either prove the presence of a transient U(II) or rule out a highly reactive U(III) intermediate resulting from the interaction of USb_2 -U with N_2 and these hypotheses should in my opinion both be presented as possible alternatives.

Once this point has been dealt with the paper is ready for publication.

RESPONSE: We take the point, and after considering the matter have not amended the exact text highlighted by the reviewer, as it didn't quite fit the flow of the prose, but slightly further down in that section we have changed a portion to read “...and the likely importance of U(II) to N_2 reduction in this context though the Sb_2^{2-} acting as an electron reservoir, for example generating an activated U(III) Sb_2 U(III) unit, may also play a role.” We feel that appropriately acknowledges the issue the reviewer correctly raises. We thank the reviewer for their support, thoughts and time – the result is an improved manuscript.